# Post-COVID conditions following COVID-19 vaccination: a retrospective matched cohort study of patients with SARS-CoV-2 infection

COVID-19 vaccinations protect against severe illness and death, but associations with post-COVID conditions (PCC) are less clear. We aimed to evaluate the association between prior COVID-19 vaccination and new-onset PCC among individuals with SARS-CoV-2 infection across eight large healthcare systems in the United States. This retrospective matched cohort study used electronic health records (EHR) from patients with SARS-CoV-2 positive tests during March 2021-February 2022. Vaccinated and unvaccinated COVID-19 cases were matched on location, test date, severity of acute infection, age, and sex. Vaccination status was ascertained using EHR and integrated data on externally administered vaccines. Adjusted relative risks (RRs) were obtained from Poisson regression. PCC was defined as a new diagnosis in one of 13 PCC categories 30 days to 6 months following a positive SARS-CoV-2 test. The study included 161,531 vaccinated COVID-19 cases and 161,531 matched unvaccinated cases. Compared to unvaccinated cases, vaccinated cases had a similar or lower risk of all PCC categories except mental health disorders (RR: 1.06, 95% CI: 1.02–1.10). Vaccination was associated with ≥10% lower risk of sensory (RR: 0.90, 0.86–0.95), circulatory (RR: 0.88, 0.83–0.94), blood and hematologic (RR: 0.79, 0.71–0.89), skin and subcutaneous (RR: 0.69, 0.66–0.72), and non-specific COVID-19 related disorders (RR: 0.53, 0.51–0.56). In general, associations were stronger at younger ages but mostly persisted regardless of SARS-CoV-2 variant period, receipt of ≥3 vs. 1–2 vaccine doses, or time since vaccination. Pre-infection vaccination was associated with reduced risk of several PCC outcomes and hence may decrease the long-term consequences of COVID-19.

Following the acute stage of illness, a broad spectrum of sequelae of SARS-CoV-2 infection has been reported in up to one-third of recovered patients[1–6], and is thought to affect more than 6% of all U.S. adults[7]. Termed 'post-COVID conditions (PCC)', these include fatigue, loss of taste or smell, shortness of breath, cough, headache, pain, and a range of moderate to severe outcomes affecting the cardiovascular, pulmonary, renal, endocrine, and neurological systems[6,8]. However, in part due to their diverse and often non-specific clinical presentation

resulting in methodological challenges when inferring causality, an exact and agreed PCC definition has not been established.

COVID-19 vaccination is effective in preventing SARS-CoV-2 infection[9], reducing disease severity[10], and preventing hospitalization or death related to COVID-19[11–13]. However, few studies have assessed the association between COVID-19 vaccination and PCC[14–18]. Of these, most had short periods of follow-up[16], or contained low numbers of vaccinated individuals as they were conducted during periods when

✉e-mail: Debbie.E.Malden@kp.org

vaccination was not widely available[18]. Robust investigations with long follow-up durations are scarce among large populations using electronic health records (EHR) across all healthcare settings[14,17].

This study aims to assess the association between prior COVID-19 vaccination status and incident PCC among individuals with SARS-CoV-2 infection enrolled at 8 large integrated healthcare systems.

## Results

### Study population

After applying exclusion and inclusion criteria, the analysis included 161,531 COVID-19 cases among patients with a documented COVID-19 vaccination at least 14 days prior to the date of test and 161,531 matched COVID-19 cases among unvaccinated patients (Table 1; Supplementary Fig. 1). The vast majority of vaccinated cases had received at least two doses of COVID-19 vaccination prior to their positive SARS-CoV-2 test ($N = 156,152/161,531$; 96.7%). Less than 1% of included COVID-19 cases had evidence of SARS-CoV-2 re-infection within the study period, and hence contributed to the analysis more than once ($N = 2297/320,725$; 0.7%).

Among all COVID-19 cases included in the analysis, the mean age was 37.1 (standard deviation [SD] 17.9) years, 54.5% ($N = 176,134$) were female. Most of the study population identified as either Hispanic (36.2%; $N = 116,984$) or White (37.1%; $N = 119,978$) race/ethnicity. Individuals of Black (8.3%; N = 26,870) and Asian (8.1%; N = 26,119) race/ethnicity each accounted for less than 10% of the study population. Most patients with SARS-CoV-2 were identified in December 2021 and January 2022, approximately reflecting the time period during which Omicron was the dominant circulating variant and COVID-19 vaccination coverage was high in the US (Supplementary Fig. 2). Compared with unvaccinated patients, vaccinated patients were more likely to be Asian (12.1% vs. 4.0%; Absolute standardized mean difference [ASMD] = 0.32) and less likely to have Medicaid subsidized insurance (11.0% vs. 16.3%; ASMD = 0.16) (Table 1). Healthcare utilization (number of outpatient or virtual encounters in the prior year) differed between vaccinated and unvaccinated patients, with a greater proportion of vaccinated patients having at least four outpatient or virtual healthcare encounters in the previous year compared with unvaccinated patients (67.8% vs. 54.0%; ASMD = 0.32). Unvaccinated COVID-19 cases also had a lower rate of influenza vaccination in the year prior to index date compared with vaccinated patients (30.1% vs. 66.7%; ASMD = 0.79) and were more likely than vaccinated cases to have a previous documented SARS-CoV-2 infection prior to their index date (10.3% vs. 6.9%; ASMD = 0.12). Most SARS-CoV-2 infections were identified via PCR, and this proportion was similar between vaccinated and unvaccinated SARS-CoV-2 positive patients (98.3% vs. 98.8%; ASMD = 0.05).

### Vaccination status and incident PCC

During a median follow-up period of 151 days, a total of 158,404 new-onset PCC outcomes were identified. In adjusted analyses, the risk of PCC was significantly lower for vaccinated vs. unvaccinated patients for 9 of the 13 PCC outcomes studied (Fig. 1). The largest reduction in relative risk (RR) associated with vaccination was observed for non-specific COVID-19 related outcomes (RR: 0.53, 95% CI: 0.51–0.56), followed by skin and subcutaneous tissue disorders (0.69, 0.66–0.72), blood and hematologic disorders (0.79, 0.71–0.89), circulatory disorders (0.88, 0.83–0.94), and sensory disorders (comprised of ear, nose, and throat disorders or visual disturbances: 0.90, 0.86–0.95). Although only marginally significant, vaccinated patients had a slightly increased risk of mental health disorders (comprised of anxiety, psychotic disorder, or depression/mood disorders) compared with unvaccinated patients over the study period (1.06, 1.02–1.10). In subgroup analyses, this positive association was most apparent in adolescents aged 12–17 years (1.12, 1.00–1.24; Fig. 2), although this association was not

statistically significant. No other PCC outcomes were associated with elevated risk following vaccination overall or by age group. Although not formally assessed, the apparent lower risk of PCC associated with vaccination appeared stronger among children compared with older age groups for the following PCC categories: gastrointestinal, symptoms (representing headache, body ache/myalgia, fever/malaise/fatigue, lymphadenopathy, weight loss, or vertigo), respiratory, sensory, and skin and subcutaneous tissue disorders. Non-specific COVID-19 related disorders were the only PCC category for which the protective effect of vaccination appeared attenuated among children compared with adults. In general, older adults (aged ≥ 65 years) had similar associations to younger adults, with greater uncertainty.

Although in general the associations were consistent across the other subgroups studied, there was some evidence of effect modification by SARS-CoV-2 variant period for a number of PCC outcomes, whereby infections occurring during the Omicron period displayed slightly enhanced associations with pre-infection vaccination status (Fig. 3). For example, risk of gastrointestinal disorders and symptoms appeared lower following vaccination during the Omicron period compared with infections occurring during the pre-Omicron period, and the positive association with mental disorders appeared to be specific to infections occurring during the Omicron period. In contrast, the observed protective effect of vaccination on skin and subcutaneous tissue disorders was enhanced during the pre-Omicron period compared with the Omicron period. Severe illness in the acute stage of SARS-CoV-2 infection appeared to modify a small number of associations, notably resulting in elevated risk associated with vaccination for renal and circulatory disorders among patients hospitalized with COVID-19, but the limited number of events resulted in wide confidence intervals (Supplementary Fig. 3). For most PCC outcomes studied, vaccination was associated with reduced risk of PCC regardless of time since last COVID-19 vaccine dose ( < 90, 90–180 or > 180 days; Supplementary Fig. 4) or whether the patient had received ≥ 3 vs. 1–2 doses of vaccine prior to their SARS-CoV-2 positive test (Supplementary Fig. 5).

### Sensitivity analysis

A total of 707,425 patients with 717,484 confirmed SARS-CoV-2 infections were included in an unmatched sensitivity analysis (Supplementary Table 1), of whom 56.1% ($N = 402,462$) had documentation of a COVID-19 vaccination at least 14 days prior to the date of SARS-CoV-2 test. In earlier periods, unvaccinated persons outnumbered vaccinated persons, and in later periods, the majority of the study population was vaccinated (Supplementary Fig. 2). Among this unmatched population, vaccinated patients were older (43.0 years [SD 18.7]) than unvaccinated patients (27.4 years [SD 20.6]) and had higher rates of comorbidities. A greater proportion of vaccinated vs. unvaccinated patients were female (57.3% vs. 51.5%, respectively), Asian (12.2% vs. 5.7%, respectively), and more likely to be vaccinated against influenza in the year prior (67.5% vs. 40.7%, respectively) (Supplementary Table 1). In adjusted analyses, the associations between vaccination status and PCC categories were similar compared with the matched analyses, despite the observed differences in study population characteristics (Supplementary Fig. 6). In analyses of PCC sub-conditions, associations were directionally concordant with the overall effect size of each PCC category, but for some rare outcomes, there was a lack of statistical power (Supplementary Fig. 7). When results were restricted to patients without underlying PCC disorders in the year prior to the positive SARS-CoV-2 test date, the associations were mostly unchanged (Supplementary Fig. 8). When follow-up was started at 90 days instead of 30 days, associations for most PCC outcomes appeared stronger (Supplementary Fig. 9). When patients were matched on influenza vaccination status in the year prior to the positive SARS-CoV-2 test date, the results were mostly unchanged (Supplementary Fig. 10).

**Table 1 | Characteristics of the study population included in the main analysis, stratified by vaccination status**

| | Unvaccinated (N = 161,531) | Vaccinated (N = 161,531) | Total (N = 323,062) | P-value* | ASMD |
|---|---|---|---|---|---|
| **Age at index date, years** | | | | 1 | 0.00 |
| < 12 | 11062 (6.8%) | 11062 (6.8%) | 22124 (6.8%) | | |
| 12–17 | 16478 (10.2%) | 16478 (10.2%) | 32956 (10.2%) | | |
| 18–24 | 16284 (10.1%) | 16284 (10.1%) | 32568 (10.1%) | | |
| 25–39 | 48567 (30.1%) | 48567 (30.1%) | 97134 (30.1%) | | |
| 40–49 | 28298 (17.5%) | 28298 (17.5%) | 56596 (17.5%) | | |
| 50–64 | 29299 (18.1%) | 29299 (18.1%) | 58598 (18.1%) | | |
| 65–74 | 7950 (4.9%) | 7950 (4.9%) | 15900 (4.9%) | | |
| 75+ | 3593 (2.2%) | 3593 (2.2%) | 7186 (2.2%) | | |
| **Mean (SD)** | 37.1 (17.9) | 37.1 (17.9) | 37.1 (17.9) | 1 | 0.00 |
| **Median [Q1, Q3]** | 36.0 [23.0, 50.0] | 36.0 [23.0,50.0] | 36.0 [23.0, 50.0] | | |
| **Sex** | | | | 1 | 0.00 |
| Female | 88067 (54.5%) | 88067 (54.5%) | 176134 (54.5%) | | |
| Male | 73464 (45.5%) | 73464 (45.5%) | 146928 (45.5%) | | |
| **Race and ethnicity** | | | | < 0.001 | 0.32 |
| Hispanic | 58507 (36.2%) | 58477 (36.2%) | 116984 (36.2%) | | |
| Asian | 6536 (4.0%) | 19583 (12.1%) | 26119 (8.1%) | | |
| Black | 16219 (10.0%) | 10651 (6.6%) | 26870 (8.3%) | | |
| White | 63899 (39.6%) | 56079 (34.7%) | 119978 (37.1%) | | |
| Multiple/Other/Unknown | 16370 (10.1%) | 16741 (10.4%) | 33111 (10.2%) | | |
| **Participating VSD site** | | | | 1 | 0.00 |
| A | 44319 (27.4%) | 44319 (27.4%) | 88638 (27.4%) | | |
| B | 6462 (4.0%) | 6462 (4.0%) | 12924 (4.0%) | | |
| C | 4365 (2.7%) | 4365 (2.7%) | 8730 (2.7%) | | |
| D | 4111 (2.5%) | 4111 (2.5%) | 8222 (2.5%) | | |
| E | 9552 (5.9%) | 9552 (5.9%) | 19104 (5.9%) | | |
| F | 1016 (0.6%) | 1016 (0.6%) | 2032 (0.6%) | | |
| G | 88483 (54.8%) | 88483 (54.8%) | 176966 (54.8%) | | |
| H | 3223 (2.0%) | 3223 (2.0%) | 6446 (2.0%) | | |
| **SARS-CoV-2 variant period** | | | | 0.783 | 0.00 |
| Alpha (Mar–Jun 2021) | 2856 (1.8%) | 2804 (1.7%) | 5660 (1.8%) | | |
| Delta (Jul–Nov 2021) | 50578 (31.3%) | 50578 (31.3%) | 101156 (31.3%) | | |
| Omicron (Dec 2021–Feb 2022) | 108097 (66.9%) | 108149 (67.0%) | 216246 (66.9%) | | |
| **Medicaid subsidized insurance** | 26340 (16.3%) | 17737 (11.0%) | 44077 (13.6%) | < 0.001 | 0.16 |
| **COVID-19 vaccine doses received**\*\* | | | | N/A | N/A |
| 1 | 0 (-) | 5379 (3.3%) | N/A | | |
| 2 | 0 (-) | 127687 (79.0%) | N/A | | |
| 3+ | 0 (-) | 28465 (17.6%) | N/A | | |
| **Time since most recent vaccine dose** | | | | N/A | N/A |
| < 90 days | 0 (-) | 50408 (31.2%) | N/A | | |
| 90–180 days | 0 (-) | 47957 (29.7%) | N/A | | |
| > 180 days | 0 (-) | 63166 (39.1%) | N/A | | |
| **Received influenza vaccine in prior 2 years** | 48645 (30.1%) | 107775 (66.7%) | 156420 (48.4%) | < 0.001 | 0.79 |
| **Prior positive SARS-CoV-2 test** | 16651 (10.3%) | 11068 (6.9%) | 27719 (8.6%) | < 0.001 | 0.12 |
| **Number of outpatient encounters in prior year** | | | | < 0.001 | 0.32 |
| 0 | 21529 (13.3%) | 9996 (6.2%) | 31525 (9.8%) | | |
| 1 – 3 | 52719 (32.6%) | 42025 (26.0%) | 94744 (29.3%) | | |
| 4 – 6 | 29484 (18.3%) | 35297 (21.9%) | 64781 (20.1%) | | |
| 7+ | 57799 (35.8%) | 74213 (45.9%) | 132012 (40.9%) | | |
| **Weighted Charlson Comorbidity Score** | | | | < 0.001 | 0.09 |
| 0 | 130848 (81.0%) | 126015 (78.0%) | 256863 (79.5%) | | |
| 1–2 | 24058 (14.9%) | 26396 (16.3%) | 50454 (15.6%) | | |
| 3 + | 6625 (4.1%) | 9120 (5.6%) | 15745 (4.9%) | | |

**Table 1 (continued) | Characteristics of the study population included in the main analysis, stratified by vaccination status**

|  | Unvaccinated (*N* = 161,531) | Vaccinated (*N* = 161,531) | Total (*N* = 323,062) | *P*-value* | ASMD |
|---|---|---|---|---|---|
| Follow-up days |  |  |  | < 0.001 | 0.44 |
| Mean (SD) | 135.9 (38.4) | 116.7 (49.0) | 126.3 (45.1) |  |  |
| Median [Q1, Q3] | 151.0 [151.0, 152.0] | 151.0 [81.0, 151.0] | 151.0 [117.0, 151.0] |  |  |

*ASMD* Absolute standardized mean difference, *N/A* Not applicable. *Differences across categories of population characteristics were compared between vaccinated and unvaccinated cases using $\chi^2$ test for categorical variables and Wilcoxon rank sum test for continuous variables. **COVID-19 vaccine count defined as number of documented mRNA COVID-19 vaccines (Pfizer-BioNTech or Moderna) or Janssen (Johnson & Johnson) COVID-19 vaccines received ≥ 14 days prior to index.

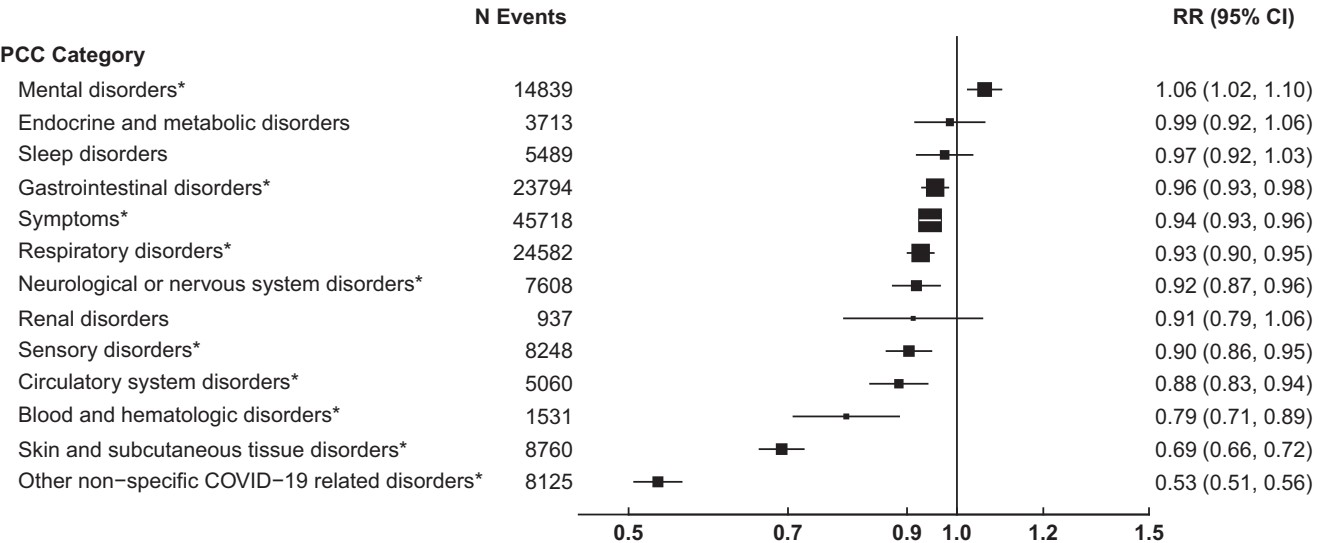

**Fig. 1 | Association of prior COVID-19 vaccination and risk of PCC categories 6 months following SARS-CoV-2 infection.** Association of prior vaccination status with Post-COVID Conditions (PCC) was estimated among 161,531 vaccinated patients matched with 161,531 unvaccinated patients on Vaccine Safety Datalink site, date of SARS-CoV-2 positive test (± 7 days), age (exact year), sex, and severity of infection (hospital admission with COVID-19 diagnosis within 7 days of SARS-CoV-2 positive test). Relative risks (RR) and 95% confidence intervals (CI) were estimated by Poisson regression adjusted for matched variables and prior SARS-CoV-2 infection, race and ethnicity, Charlson comorbidity score, Medicaid status, influenza vaccination, and healthcare utilization in the year prior. Box sizes are inverse-variance weighted. PCC category 'symptoms' included headache, body ache/myalgia, fever/malaise/fatigue, lymphadenopathy, weight loss, or vertigo. To limit the impact of multiple testing, Bonferroni correction was applied to main analysis of 13 PCC categories, with 2-sided *p*-values at a level of significance of 0.004. *Significant with Bonferroni correction.

## Discussion
### Main findings
Using EHR data from 8 large US integrated healthcare systems, the current study demonstrated a reduction in the incidence of most PCC outcomes associated with COVID-19 vaccination among more than 300,000 patients with SARS-CoV-2 infection. In general, associations between vaccination status and PCC persisted across age groups, although with some evidence for slightly stronger associations at younger ages. Furthermore, most associations between vaccination and PCC persisted across SARS-CoV-2 variant periods studied (i.e., pre-Omicron vs. Omicron) and appeared to be unaffected by receipt of a third vaccine dose or time between last vaccination and SARS-CoV-2 infection.

### Comparison with prior literature
Recent systematic reviews investigating the impact of prior vaccination on PCC have demonstrated an overall protective effect of vaccination[16,19], although reported associations from previous observational studies have varied greatly between studies[2,14,15,20,21]. However, there is a general lack of robust large-scale longitudinal studies describing associations between vaccination status and PCC. Moreover, despite recent work to improve the definition of PCC[22], outcomes used in earlier studies differ due to a lack of consensus on PCC definitions at the time of writing, making comparisons challenging.

The most robust evidence to date is from large-scale integrated healthcare systems like the current study[17], that have the advantage of standardized EHR data to capture PCC outcomes across all healthcare settings among large populations. The largest EHR-based study assessing the impact of prior vaccination on PCC outcomes prior to the current study included 33,940 fully vaccinated persons from a US national healthcare database of mostly adults aged over 65 years[14]. By comparing PCC risk with unvaccinated comparators, the authors observed a similar protective effect of vaccination against PCC, akin to the findings in the current study, with an overall reduction in risk of approximately 15%. Also, although a slightly different definition of PCC was used, they observed a particularly strong protective effect of vaccination against selected PCC categories similar to the current study, such as hematologic and respiratory disorders. Another large study assessing the impact of pre-infection vaccination on PCC using two distinct definitions of PCC - either a specific long COVID-19 clinical diagnosis (*N* = 47,404) or a previously described computational phenotype (*N* = 198,514) - found similar reduced odds of PCC among vaccinated patients compared with unvaccinated patients[23]. Other prior EHR-based studies conducted in the US, UK, and France have also demonstrated that vaccinated patients are less likely to present with long-term COVID-19 symptoms compared with unvaccinated patients[1,15,17,18,24,25]. With complete EHR data on more than 300,000 patients from integrated healthcare systems in the US, the current study represents the largest cohort study describing the association

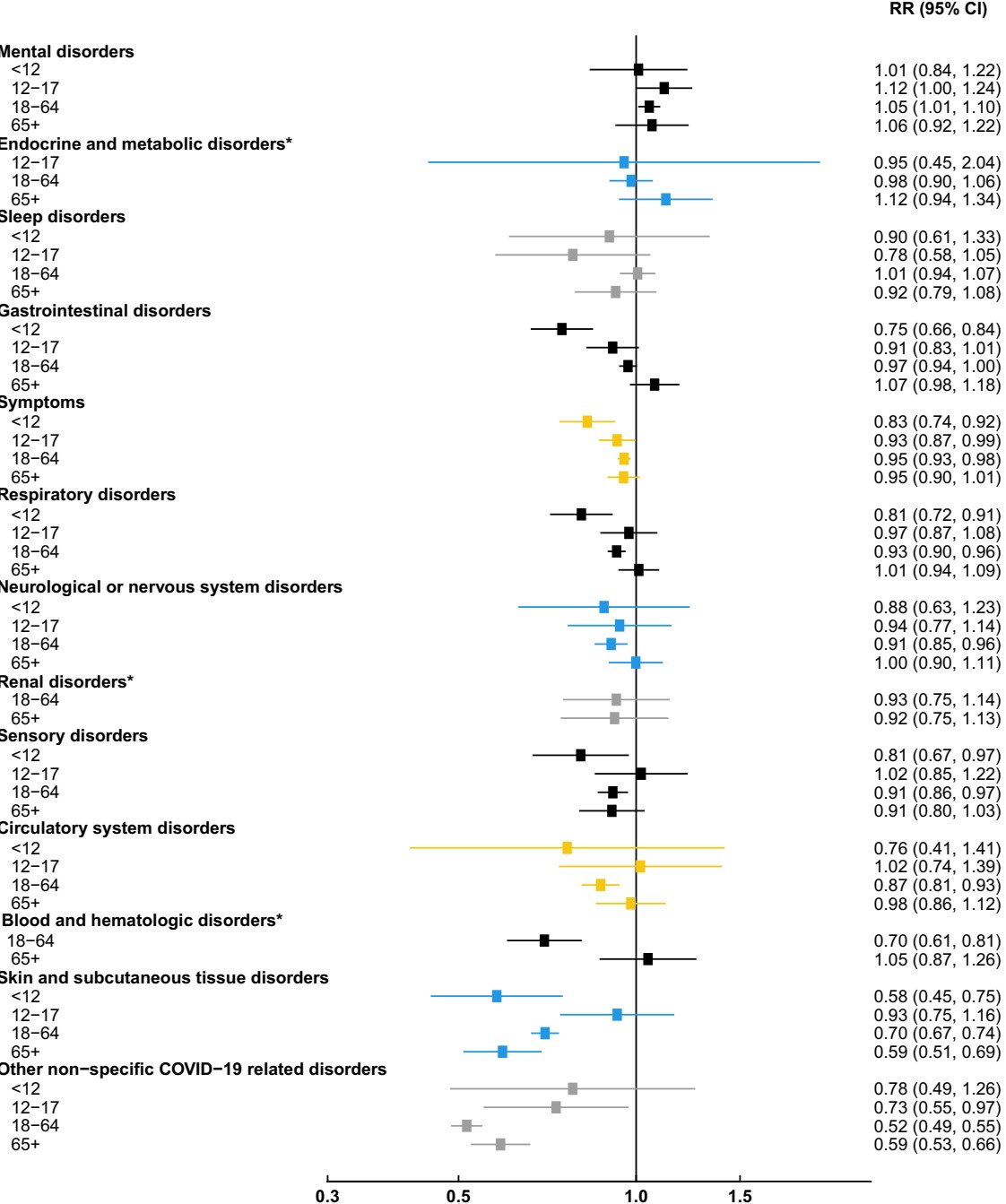

**Fig. 2 | Associations of prior COVID-19 vaccination and risk of PCC categories 6 months following SARS-CoV-2 infection, by age group.** Association of prior vaccination status with Post-COVID Conditions (PCC) was estimated among 161,531 vaccinated patients matched with 161,531 unvaccinated patients on Vaccine Safety Datalink site, date of SARS-CoV-2 positive test (± 7 days), age (exact year), sex, and severity of infection (hospital admission with COVID-19 diagnosis within 7 days of SARS-CoV-2 positive test). Relative risks (RR) and 95% confidence intervals (CI) were estimated by Poisson regression adjusted for matched variables and prior SARS-CoV-2 infection, race and ethnicity, Charlson comorbidity score, Medicaid status,

influenza vaccination, and healthcare utilization in the year prior. PCC category 'symptoms' included headache, body ache/myalgia, fever/malaise/fatigue, lymphadenopathy, weight loss, or vertigo. *Associations not shown due to lack of events for PCC outcomes in the following age ranges: Endocrine and metabolic disorders among < 12 years (RR: 1.51, 95% CI: 0.30–7.61); Renal disorders among < 12 years (no estimate, due to lack of events) and 12–17 years (RR: 1.31, 95% CI: 0.06–26.71); Blood and hematologic disorders among < 12 years (no estimate, due to lack of events) and 12–17 years (RR: 0.73, 95% CI: 0.26–2.08).

between vaccination status and PCC categories to date. Therefore, whereas others have lacked statistical power to assess some PCC outcomes that are particularly concerning to patients and clinicians such as acute respiratory disorders, cardiac arrythmias/postural orthostatic tachycardia syndrome (POTS), neurological disorders, and sensory disorders[17], our study was able to confirm a strong and persistent protective effect of COVID-19 vaccination.

In our study, while the observed protective effect of COVID-19 vaccination against PCC generally persisted across all age groups, some associations were more pronounced among younger populations, as has been shown elsewhere[17]. While vaccination has been shown to decrease the severity of illness among persons aged < 18 years previously[26], the current study is the first large-scale study to investigate the association between prior COVID-19 vaccination and

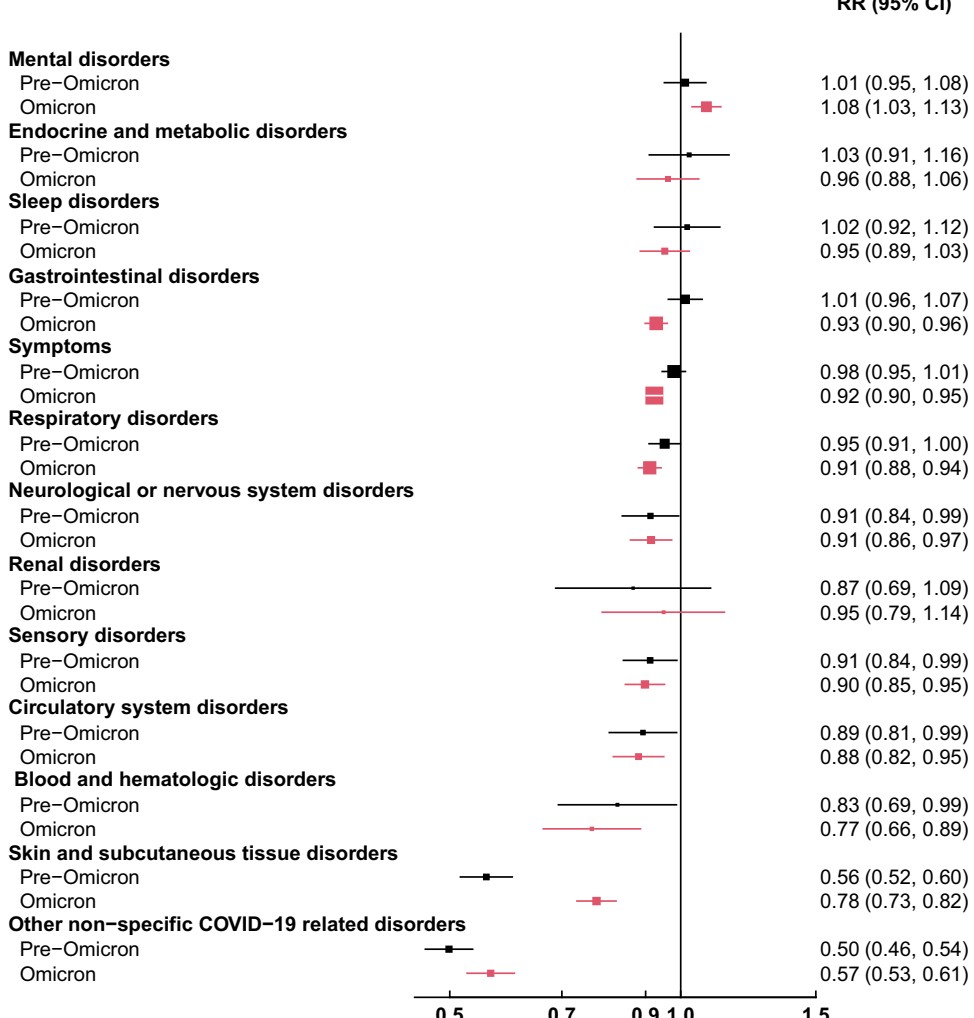

**Fig. 3 | Associations of prior COVID-19 vaccination and risk of PCC categories 6 months following SARS-CoV-2 infection, by SARS-CoV-2 variant period\*.** Association of prior vaccination status with Post-COVID Conditions (PCC) was estimated among 161,531 vaccinated patients matched with 161,531 unvaccinated patients on Vaccine Safety Datalink site, date of SARS-CoV-2 positive test (± 7 days), age (exact year), sex, and severity of infection (hospital admission with COVID-19 diagnosis within 7 days of SARS-CoV-2 positive test). Relative risks (RR) and 95% confidence intervals (CI) were estimated by Poisson regression adjusted for matched variables and prior SARS-CoV-2 infection, race and ethnicity, Charlson comorbidity score, Medicaid status, influenza vaccination, and healthcare utilization in the year prior. Box sizes are inverse-variance weighted. PCC category 'symptoms' included headache, body ache/myalgia, fever/malaise/fatigue, lymphadenopathy, weight loss, or vertigo. \*Pre-Omicron: March 1st 2021 – November 30th 2021; Omicron: December 1st 2021 – February 28th 2022.

PCC among children and adolescents. Hence, the protective effect of COVID-19 vaccination against PCC among children and adolescents is an important finding. However, although associations with PCC may be stronger, in general, PCC outcomes among younger ages occur less frequently overall[27]. Interestingly, the association with mental disorders and prior COVID-19 vaccination appeared dominated by adolescents in the current study. Importantly, mental health outcomes could reflect general health seeking behaviors more broadly[28,29], and hence may be closely correlated with vaccine uptake perhaps to a greater extent than other PCCs. In further support of this, the association with mental health disorders was dominated by infections occurring during Omicron period, during which vaccination uptake was high. Hence, persons who actively remained unvaccinated (i.e., 'vaccine hesitant' populations) during this period may have differed systematically with respect to general healthcare seeking behavior compared with unvaccinated cohorts during earlier study periods.

To date, the mechanisms underlying the protective effect of vaccination on PCC among breakthrough infections have not been fully elucidated. A conceivable protective effect of the vaccine on PCC could be attributed to less severe COVID-19 illness, which itself is a predictor of PCC[30]. It is possible that the abnormalities observed across multiple organ systems with PCC are mitigated by the vaccine through lowering viral reservoirs or reducing the inflammatory and/or immune responses often associated with the PCC syndrome[31]. The observed positive association between COVID-19 vaccination and mental health disorders has not been clearly elucidated. However, in addition to the potential confounding by healthcare seeking behavior described above, the association could be due to stress and anxiety induced by experiencing breakthrough SARS-CoV-2 infection despite being vaccinated. Importantly, the main protective effect of COVID-19 vaccination on PCC is likely exerted through the prevention of the primary SARS-CoV-2 infection. Although this was not assessed in the current study, it is an important additional benefit to consider when assessing the overall value of COVID-19 vaccination.

### Potential limitations
There are potential limitations to this study. First, vaccinated and unvaccinated individuals likely differ with respect to lifestyle, demographic and clinical factors – many of which are difficult to account for in a real-world analysis and are themselves tightly correlated with

comorbidities such as PCC outcomes. This is a common limitation that is encountered ubiquitously across many retrospective cohort studies. Our study population was restricted to individuals with a documented SARS-CoV-2 positive test result, which helps to mitigate some of the concerns around differences in care-seeking behavior between vaccinated and unvaccinated individuals. SARS-CoV-2 tests were widely available during the study period and therefore testing behaviors should not have differed by severity of illness or vaccination status. Furthermore, we have matched vaccinated and unvaccinated individuals on important confounding variables such as age, sex, time and severity of SARS-CoV-2 infection. We also adjust for comorbidity status, healthcare seeking behavior, and ethnicity to reduce the impact of confounding. In a sensitivity analysis excluding all individuals with prior documentation of each PCC outcome in the year prior, associations were mostly unchanged (Supplementary Fig. 8), indicating that any potential bias related to prior disease did not substantially alter the associations. Second, as with all vaccine-related studies, misclassification of vaccination status was possible, but an advantage of conducting this study within the Vaccine Safety Datalink (VSD) network of integrated healthcare systems was that the scale of this misclassification was likely small due to concerted efforts to collect complete records, including through external providers[32]. Third, the analysis covers a time period when infections were dominated by the Omicron variant, but the PCC definition was based on studies conducted among patients infected with prior circulating SARS-CoV-2 variants[15]. However, while PCC may present differently by variant[33], the conditions included in our PCC definition were broad and it was unlikely that important conditions were omitted. Further, in an effort to enhance the comparability of our findings, we used a PCC definition that has been used consistently by CDC in prior publications[34–36]. However, it must be noted that other sources have used narrower definitions of PCC, such as the World Health Organization and the National Institute for Health and Care Excellence (NICE). For comparability, we investigated these alternative PCC definitions in a sensitivity analysis and associations were directionally concordant with our overall PCC categories, with less precision since EHR capture may be incomplete for the mild symptoms that are unlikely to result in healthcare utilization (Supplementary Fig. 11). Furthermore, others also define PCC as persistent symptoms beyond 12 weeks of infection, which is much longer than the current study. However, we observed stronger associations in a sensitivity analysis using this definition (Supplementary Fig. 9), reinforcing the primary finding that vaccination appears protective against long-term sequelae of COVID-19. Therefore, given the directional concordance of these associations, we feel that the 30-day PCC definition used in the current study is appropriate and maximizes statistical power. Fourth, the current study did not account for the use of COVID-19 treatments, which may be associated with vaccination status, disease severity, and PCC outcomes. However, effective antiviral drugs for the treatment of mild-to-moderate COVID-19 were not widely prescribed during the study period, and therefore the impact on associations was likely minimal. In addition, we did not use medication, procedure codes, or NLP-supplemented analysis to identify additional events[14,18,37], which may have increased statistical power. However, due to the large study population, there was sufficient power to assess associations for most analyses using a matched cohort design. In addition, while the inclusion of children was a particular strength of our study, one challenge when assessing PCC risk in this population was the absence of consistent information on PCC definitions, since children are understudied and pediatric PCC may present differently compared with adults[38]. Therefore, PCC associations among children should be interpreted with caution, particularly for those aged under 12 years. This uncertainty may partially explain the observed association between vaccination status and increased risk of mental disorders among children in the current study, which was not associated with SARS-CoV-2 infection among children in a previous

analysis of VSD data[35]. Furthermore, while matching on index date and important demographic or clinical factors reduced confounding bias, it is possible matching could have introduced bias if excluded individuals differed with respect to PCC. However, unmatched sensitivity analyses displayed similar associations overall despite clear differences between vaccinated and unvaccinated populations (Supplementary Fig. 6). Also, PCC was grouped into categories based on mechanistic and pathophysiological similarities, which may have obscured associations for individual sub-conditions. However, in general, associations with sub-conditions were directionally concordant with the overall effect size for each PCC category (Supplementary Fig. 7). Lastly, the VSD population is comprised of individuals with health insurance coverage, and hence the findings may not be generalizable to uninsured populations. However, despite this, the VSD population appears to have similar demographic characteristics to the catchment areas served by the respective health plans[39,40].

In summary, our findings provide evidence for a reduction in most new-onset PCC associated with prior COVID-19 vaccination prior to and during the Omicron wave in the US. Moreover, this apparent protective effect of prior vaccination persisted regardless of age, receipt of ≥ 3 vs. 1–2 vaccine doses, or time since vaccination. While an association was observed between vaccination status and increased risk of mental disorders, further research is warranted to investigate this finding and possible explanations other than vaccination. As new vaccines are developed and further variants arise, ongoing research will be important to evaluate associations in risk of PCC.

## Methods

This study was reviewed and approved by the institutional review boards of all participating health care organization sites (reference ID: 4982) with a waiver of informed consent and was conducted consistent with federal law and CDC policy. See, for example 45 C.F.R. part 46.101(c); 21 C.F.R. part 56.

### Setting and study population

The current study was conducted within the VSD network, a research network of eight integrated healthcare systems in the United States (US) that enables comprehensive analysis of EHR data from inpatient, emergency, outpatient, and virtual care settings[41]. In addition, data on externally administered COVID-19 vaccinations were integrated into the EHR[32]. The study population included patients of all ages enrolled at a VSD site with at least one documented SARS-CoV-2 positive test result (PCR or antigen) from March 1st, 2021, to February 28th, 2022, and 1-year continuous health plan membership (allowing for a 31-day administrative gap) prior to their SARS-CoV-2 positive test date (index date). To ensure that patients were active members at the start of follow-up (beginning 30 days following the index date), membership was also required for the 30 days after the index date. The analysis was conducted at the SARS-CoV-2 infection level. Therefore, multiple positive tests per patient were included if they occurred at least 90 days apart. Multiple SARS-CoV-2 test results within 90 days were considered as the same infection and the first test result was used as the index date in the analysis[42]. To ensure sufficient time for immunological response following vaccination and to accurately distinguish between PCC and post-vaccine reactions, patients were excluded if they received a COVID-19 vaccine within 14 days prior to or within 30 days after their SARS-CoV-2 positive test date. Patients were also excluded if they were missing information on age or sex since these were important adjustment variables. In addition, if patients received either a Janssen (Johnson & Johnson) COVID-19 vaccine or a COVID-19 vaccination not routinely administered in the US, they were not included in the analysis since it is likely that these patients systematically differ from mRNA vaccine recipients and the limited sample size would preclude meaningful subgroup analysis among these individuals.

## Exposure

The main exposure of interest was COVID-19 vaccination, defined as EHR documentation of an FDA-authorized COVID-19 vaccine (Pfizer-BioNTech or Moderna mRNA COVID-19 vaccines) at least 14 days prior to the documented SARS-CoV-2 positive test occurring over the study period, regardless of vaccine dose. Unvaccinated comparators included COVID-19 cases without a documented COVID-19 vaccination at least 14 days prior to the date of the SARS-CoV-2 positive test.

## Outcome

We derived our PCC definition based on a CDC definition developed in 2021 used in an earlier analysis of VSD data[35]. However, to reduce the number of PCC outcomes studied, we defined 13 broader PCC categories consisting of 51 sub-conditions according to affected organ system, as has been done elsewhere (Supplementary Table 2)[6,43]. Individual PCC sub-conditions were defined according to a pre-specified list of diagnosis codes (International Classification of Diseases, 10th Revision [ICD-10] codes). The outcome was defined as any documentation of new-onset pre-specified PCC outcomes occurring $\geq 30$ days after the SARS-CoV-2 positive test (i.e., following the acute stage of infection). To determine new-onset PCC status, pre-existing PCCs were identified during outcome-specific look-back periods (Supplementary Table 2). If a pre-existing PCC event was identified within this look-back period, re-occurrence within the follow-up period did not contribute to the analyses. For example, if an individual in the study had diabetes mellitus documented in their EHR within twelve months prior to the index date, diabetes mellitus codes identified during the study follow-up period were not identified as a new-onset PCC outcome in the analysis. This approach was favored over the exclusion of individuals with pre-existing PCC events prior to the study period because it ensured that the study population was consistent across all PCC events studied.

## Follow up

All individuals in the study were followed from 30 days to 6 months after the date of positive SARS-CoV-2 test (i.e., index date). When patients had a valid documented SARS-CoV-2 re-infection (i.e., SARS-CoV-2 positive test occurring at least 90 days from the previous SARS-CoV-2 positive test date) occurring over the study period, follow-up was censored at the date of re-infection and re-started 30 days following re-infection. Otherwise, patients were censored at the date of receipt of an additional dose of COVID-19 vaccine (or first dose, if previously unvaccinated), termination of health plan membership, or death (whichever occurred first).

## Statistical analysis

Population characteristics were described using mean and standard deviation for continuous variables and frequency and percentage for categorical variables. Differences between vaccinated and unvaccinated groups were compared using an independent $t$ test or chi-square ($\chi^2$) test. Absolute standardized mean differences (ASMD) were calculated to assess the balance of covariates between exposure groups.

For the main analyses, vaccinated individuals with SARS-CoV-2 positive tests were matched (1:1) with unvaccinated individuals with SARS-CoV-2 positive tests on VSD site (8 sites), date of SARS-CoV-2 positive test (+/−7 days), age (exact age in years), sex (male, female), and severity of infection (hospital admission with COVID-19 diagnosis within 7 days of SARS-CoV-2 positive test). Poisson regression models were used to estimate relative risks (RR) and corresponding 95% confidence intervals (CI) for the association between vaccination status and PCC categories, with robust variance (generalized estimating equation models) to account for correlation between repeated infections within the same individual. Covariates in the multivariable models were derived from EHR data and included matching variables, previous SARS-CoV-2 infection (SARS-CoV-2 positive test occurring at least 90 days prior to the index date,

including those occurring prior to and during the study period), healthcare utilization in the year prior to index date (number of outpatient and virtual visits), influenza vaccination in the 2 years prior (yes/no), Medicaid insurance status, race and ethnicity (five mutually exclusive categories: Hispanic, Black, Asian, White, and Other/Unknown), and weighted Charlson comorbidity score[44,45] in the year prior to the index date.

Effect modification by age ( < 12, 12-17, 18-64, ≥65 years), SARS-CoV-2 variant period (pre-Omicron: March 1st 2021 – November 30th 2021; Omicron: December 1st 2021 – February 28th 2022), and severity of the acute SARS-CoV-2 infection (hospitalization with COVID-19 diagnosis within 7 days of index date, yes/no) was assessed using sub-group analyses by fitting separate models within these pre-specified strata. Vaccine effect modification was also assessed by time since most recent dose of vaccine ( < 90, 90-180, >180 days) and number of vaccine doses received prior to the index date (1-2, ≥3 doses). To assess the robustness of estimates among the total population of all SARS-CoV-2 positive patients over the study period, we conducted an unmatched sensitivity analysis with the same covariates included in the multivariable models as those listed above. To assess directional concordance of effect estimates for sub-conditions with the overall effect size estimates for PCC categories, we repeated the main analysis for all 51 sub-conditions. As an additional sensitivity analysis, we excluded all persons with each PCC documented in their EHR in the year prior to the date of positive SARS-CoV-2 test. For comparability with other studies using distal time periods to define PCC relative to the initial SARS-CoV-2 positive test date, we repeated the primary analysis using a follow up period of 90 days – 6 months. In addition, we performed a sensitivity analysis to assess the impact of potential vaccine hesitancy and general healthcare seeking behavior on the observed association with PCC by matching on influenza vaccination status in the year prior to the positive SARS-CoV-2 test. To limit the impact of type I error, Bonferroni correction was applied to the main analysis, with 2-sided p-values at a level of significance of 0.004 (calculated as 0.05/13). All statistical analyses were performed using SAS statistical software version 9.4 (SAS Institute, Cary, NC) and all graphics were developed using R version 4.0.5.

## Reporting summary

Further information on research design is available in the Nature Portfolio Reporting Summary linked to this article.

## Data availability

The data that support the study conclusions are unavailable for public access. Guidelines on how to access VSD data through a sharing program administered by the National Center for Health Statistics Research Data Center (NCHSRDC) are provided here: https://www.cdc.gov/vaccinesafety/ensuringsafety/monitoring/vsd/data-sharing-guidelines.html and are subject to change.

## Code availability

Statistical code is available for interested readers by emailing Debbie Malden at Debbie.e.malden@kp.org.

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

## Acknowledgements

We would like to thank Sudha Medalbalimi, MS (Marshfield Clinic Research Institute), Rachael Doud (KPWHRI), Matthew Slaughter, MS (Center for Health Research, Kaiser Permanente Northwest), and Bradley Crane, MS (Center for Health Research, Kaiser Permanente Northwest) for their contributions to data collection and preparation. We thank Hannah Berger, MPH (Marshfield Clinic Research Institute), Kayla Hanson, MPH (Marshfield Clinic Research Institute), Kristin Goddard (Kaiser Permanente Northern California, Vaccine Study Center), and Erika Kiniry (KPWHRI) for their contributions to overall project management. This study was supported by the Centers for Disease Control and Prevention through the Vaccine Safety Datalink under contract 75D30122D15429. The findings and conclusions in this article are those of the authors and do not necessarily represent the official position of the Centers for Disease Control and Prevention.

## Author contributions

D.E.M constructed the analysis plan, assisted with data analysis and drafted the manuscript. D.E.M., S.Y.T., B.J.L., I.A.L., L.Q. and L.S.S. designed the project. I.A.L., C.B. and L.Q. completed the programming and data analysis. D.E.M., S.Y.T., B.J.L., D.S.R., D.T.A., I.A.L., L.Q., R.K., M.F.D., J.T.B.W., L.S.S., J.C.N., D.L.M., O.Z., M.L.H., C.C.F., E.S.W. and S.S. revised the manuscript and provided critical input.

## Competing interests

L.S.S. reports research support from Moderna for a COVID-19 vaccine effectiveness study and GlaxoSmithKline and Dynavax for unrelated studies. L.Q. reports research support from Moderna for a COVID-19 vaccine effectiveness study and GlaxoSmithKline and Dynavax for unrelated studies. S.Y.T. reports research support from Pfizer paid directly to institution for COVID-19 vaccine effectiveness and Paxlovid studies. C.C.F. reports research support from Pfizer and Johnson & Johnson for unrelated studies. I.A.L, B.J.L., C.E.B., D.L.M., D.T.A., D.E.M., D.S.R., E.S.W., J.C.N., J.T.W, M.F.D., M.L.H., R.K., S.S. and O.Z. have no conflict of interest to disclose.

## Additional information

Debbie E. Malden [1,13] ✉, In-Lu Amy Liu[1], Lei Qian[1], Lina S. Sy [1], Bruno J. Lewin [1,2], Dawn T. Asamura[1], Denison S. Ryan [1], Cassandra Bezi[1], Joshua T. B. Williams[3], Robyn Kaiser[4], Matthew F. Daley [5], Jennifer C. Nelson[6], David L. McClure[7], Ousseny Zerbo [8], Michelle L. Henninger[9], Candace C. Fuller [10], Eric S. Weintraub[11], Sharon Saydah [11] & Sara Y. Tartof [1,12,13]

[1]Kaiser Permanente Southern California, Department of Research & Evaluation, Pasadena, USA. [2]Kaiser Permanente Bernard J. Tyson School of Medicine, Department of Clinical Science, Pasadena, USA. [3]Denver Health, Ambulatory Care Services & Center for Health Systems Research, Denver, USA. [4]HealthPartners Institute, Bloomington, USA. [5]Kaiser Permanente Colorado, Institute for Health Research, Aurora, USA. [6]Kaiser Permanente Washington Health Research Institute (KPWHRI), Seattle, USA. [7]Marshfield Clinic Research Institute, Marshfield, USA. [8]Kaiser Permanente Northern California, Division of Research, Vaccine Study Center, Oakland, USA. [9]Kaiser Permanente Northwest, Center for Health Research, Portland, USA. [10]Harvard Pilgrim Health Care Institute, Boston, USA. [11]Centers for Disease Control and Prevention, Immunization Safety Office, Atlanta, GA, USA. [12]Kaiser Permanente Bernard J. Tyson School of Medicine, Department of Health Systems Science, Pasadena, USA. [13]These authors contributed equally: Debbie E. Malden, Sara Y. Tartof. ✉e-mail: Debbie.E.Malden@kp.org

