## [Peer Review File · Nature Communications]

Post-COVID conditions following COVID-19 vaccination: A retrospective matched cohort study of patients with SARS-CoV-2 infectionEditorial Note: Parts of this Peer Review File have been redacted as indicated to remove third-party material where no permission to publish could be obtained.

REVIEWER COMMENTS

Reviewer #1 (Remarks to the Author):

I appreciate the opportunity to review this paper. It has several commendable aspects, such as longitudinal data, large sample, important dimensions were measured, and an important research question. Although there are many things to appreciate with this study, I have several concerns.

Primary concerns

1. Outcome definition - time window

You used 30 days to 6 months as the time window. The WHO definition of the post-covid condition is based on symptoms still present after 3 months from the onset of COVID-19 (i.e., 90+ day). Why not use this definition, as this is the official WHO definition? I think the main results should be tested against this definition of post-covid for robustness.

(Additionally, the strongest associations were found for "Other non-specific COVID-19 related disorders" (Fig. 1), which according to Appendix A seem like diagnoses partly describing the original COVID-19. This makes me wonder if these associations survive if starting to classify as post-covid at 90+ days.)

2. Outcome definition – symptoms included and symptom grouping

A stronger argument must be given these diagnoses, and groupings of diagnoses, should be used to characterize post-covid. According to the WHO, the most common symptoms of post-covid are (i) fatigue, (ii) shortness of breath (or difficulty breathing), (iii) memory, concentration or sleep problems, (iv) persistent cough, (v) chest pain, (vi) trouble speaking, (vii) muscle aches, (viii) loss of smell or taste, (ix) depression or anxiety, and (x) fever. What does results look like if you use these as you PCC categories? Now fatigue, which together with respiratory problems are considered the main symptoms, are grouped under "symptoms" – why? And, why did fatigue (and some other complaints) only have a 30d look back period? A stronger argument should be made for being inconsistent across diagnoses in terms of lookback period.

3. Statistical analyses – controlling for prior healthcare utilization

There are several studies suggesting that some cases of post-covid could have a considerable psychological mechanism to it, perhaps related to health anxiety. I think the analyses would be more robust if the authors also presented a version where they controlled for whether or not the individual in the previous year had a psychological diagnosis. Table 1 provides clear indications that there are significant differences between the groups in the year prior to infection. This is an indication that there still may be substantial confounding lurking in the background.

Minor issues

- line 254: "substantial reduction"? For the most well-known PCC symptoms (Fig 1), the RRs are above 0.9. I think "substantial" feels like overselling. I suggest you tone this down. Same goes for "clear evidence" (line 369).

- line 299: "the enhanced protective effect of COVID-19 vaccination against PCC among children and adolescents is a novel and important finding." This also comes across as a bit overselling. It looked like (cf. Fig 2) there only were two (gastro and respiratory) out of 13 types of symptoms where there was a significant difference. Or did I misread your figure?

- Subgroup analysis: you chose age and virus type to be included in your main subgroup analysis – what was the rationale? A short motivation for these would be helpful.

- It would be helpful with a table showing the number of events for each of the diagnoses underlying Figure 1.

- In the unmatched analysis, the RR for mental disorders is among your strongest associations (fairly strong association and tight Cis). This makes me a bit worried about confounding, although attenuated by the matching. How do you interpret this? Are there additional analysis you could do to investigate where this comes from? To speculate: it seems like those taking the vaccine have higher levels of mental health issues, and the vaccine could be one way to reduce health anxiety related to being infected by a fairly unknown, and potentially dangerous, virus.

- Line 66: I think you should be more specific in stating the aims of the study.

Reviewer #2 (Remarks to the Author):

In this study, the authors sought to assess whether the incidence of post-COVID condition differed between those with post-vaccination SARS-CoV-2 infections and those without prior vaccination at the time of infection.

This question has been addressed in several prior studies that are referenced in the introduction. The main advantage of the present study compared to existing studies is the increased sample size. However, it unfortunately comes with several major sources of bias (not present in previous studies) which cannot be addressed by increasing sample size (indeed increasing sample size only makes bias more apparent). These include:

1) Censoring for death without accounting for death as a competing risk: people who are vaccinated are less likely to die and thus more likely to develop any of the other outcomes, thus biasing all risks towards higher values.

2) Including individuals in the cohort who can never develop the outcome (i.e. those with the outcome before the exposure). The justification provided by the authors (to "ensure that the study population was consistent across all PCC events studied") is unsatisfactory as it leads to biased estimates of risks.

3) The use of "censoring" but without any time-to-event analysis. Risk ratios calculated after excluding censored individuals can be biased. To illustrate this point, imagine that Group A and Group B have 1000 people each at the beginning of a time window and both groups have 100 events during that time window, but fewer members of Group B are censored during that time window (e.g. 150 vs 50 in Group A). The risk ratio would be 1.12 ($= (100/850)/(100/950)$) even though both groups have the same number of events and same number of people at risk at the beginning of the time window.

4) The lack of any adjustment for individual comorbidities. Only adjustment for Charlson Comorbidity Score is used but individual comorbidities might still be unbalanced between cohorts, and different comorbidities might have different associations with the outcomes.

5) The lack of matching for important covariates with influences on outcomes that might be greater than the influence of age or sex, such as prior influenza vaccination which is reported to be 30.1% vs. 66.7% in the two cohorts, and likely captures vaccine hesitancy.

The increased statistical power offered by the larger sample size is also underutilised. The authors report that they are able to examine rarer outcomes, such as "acute respiratory disorders, cardiac arrhythmias/postural orthostatic tachycardia syndrome (POTS), neurological disorders, and sensory disorders" but most of these outcomes are already investigated in previous studies including reference [15] and [17].

Finally, to demonstrate the coverage of the dataset, it would be informative to report the overall vaccination coverage in the whole database (i.e. including in those who did not have COVID-19) and compare this to vaccination coverage in the American population during the same period.

Reviewer #3 (Remarks to the Author):

REVIEWER COMMENTS

Overall – well conducted study which makes an important contribution to the field with a robust study design and large sample size. Specifically, the study supports previous research regarding the positive effects of vaccination on persistent symptoms following SARS-CoV-2 infection and goes beyond existing studies by exploring this relationship in children and adolescents.

Specific comments below.

Introduction

1. The authors use PCC but other terms are in use, including Long COVID, post-covid syndrome. Please briefly describe the differences and comment as to why PCC is used in this study.
2. Please add a bit more context in terms of when and how vaccination was rolled out in US or in the geographical area covered by VSD.
3. Also please briefly explain the SARS-CoV-2 testing policy (free, universal, mandatory, etc....) during the study period as this information would give the reader an indication of the likelihood of missed infections through lack of testing. Could testing behaviour or test availability/policy influence presence or absence of a SARS-CoV-2 test result in EHR? How is this likely to affect the findings – please briefly discuss in the section of the discussion on limitations.
4. It would be helpful to provide greater context from other studies which have done similar analyses (principally those mentioned). The discussion refers to these studies but further consider the similarities and differences between the presented work and existing analyses.

Methods

1. Regarding the VSD and eight integrated healthcare systems. Can you give more detail in terms of geographical area covered and sociodemographic profile of underlying population and any information regarding quality/completeness of data in VSD or validation studies on this dataset.
2. Regarding sentence "Therefore, multiple positive tests per patient were included if they occurred at least 90 days apart." Could you add a reference to justify this assumption. Consider adding, for example reference "UK Government. Investigation and management of suspected SARS-CoV-2 reinfections: a guide for clinicians and infection specialists 2021. Available at: <https://www.gov.uk/government/publications/covid-19-investigation-and-management-of-suspected-sars-cov-2-reinfections/investigation-and-management-of-suspected-sars-cov-2-reinfections-a-guide-for-clinicians-and-infection-specialists>."
3. Patients who received either a Janssen COVID-19 vaccine or a COVID-19 vaccine not routinely administered in the US were not included in the analysis. Were there sufficient numbers to do a sub analysis by type of vaccine? Might be of interest for countries using other vaccine types.
4. WHO and NICE in UK, for example, use different definitions, for example Long COVID, post covid syndrome, persistent symptoms for ≥ 12 weeks. As a sensitivity analysis (and perhaps to be included as supplementary material) could the main analysis be replicated using at least one other alternative definition to see if the findings are consistent.
5. Regarding "weighted Charlson comorbidity score" - Could you provide a reference for this or more detail of this score in the text or supplementary material, including how it is calculated and its interpretation.

Discussion

1. From your analysis can you say anything specific about "brain fog" or memory or cognitive symptoms which have been widely reported in those with persistent symptoms following covid?
2. Comment on potential (if any) misclassification of infection status, as per comments above regarding quality/completeness of dataset (VSD) and testing practices.
3. The conclusions appear overall well supported by the work, but it would be interesting to understand better the relationship between race/ethnicity and profile of EHR population (compared to underlying overall population) to disentangle what is a bias in EHR vs what is related to COVID-19 vaccination and PCC.

Table 1 - Please give exact p values throughout.

Figures 1-3 - Figures 1,2,3 the category "symptoms" what organ/system does this refer to? Or is it non-specific symptoms - please clarify in text or in a footnote.

January 2024

Ref: Response to reviewer's comments (NCOMMS-23-49571-T)

Dear Sir/Madam,

Thank you for your time and consideration reviewing our manuscript titled 'Post-COVID conditions following COVID-19 vaccination: A matched analysis of patients with SARS-CoV-2 from 8 large integrated healthcare systems' submitted to *Nature Communications*. On behalf of all coauthors, we appreciate your insightful comments on this manuscript, and we have been able to incorporate changes to reflect most of these suggestions.

Where appropriate, we have highlighted the changes made in response to your comments and suggestions in the enclosed revised version of our manuscript. In addition, please find a point-by-point response to specific comments and concerns outlined below in blue font.

Reviewer 1

General comments

I appreciate the opportunity to review this paper. It has several commendable aspects, such as longitudinal data, large sample, important dimensions were measured, and an important research question. Although there are many things to appreciate with this study, I have several concerns.

Thank you for this comment. We agree that our study has many strengths which set it apart as a robust and unique analysis to address this important research question. However, we also understand your concerns, some of which are ubiquitous in epidemiological research and hence unavoidable. We have tried our best to mitigate where these concerns could have potentially biased our results. Otherwise, if this was not possible, we have attempted to identify the direction of any potential bias and explain any consequences on the interpretation of findings in the limitations section.

Specific comments

Comment 1: Outcome definition - time window

You used 30 days to 6 months as the time window. The WHO definition of the post-covid condition is based on symptoms still present after 3 months from the onset of COVID-19 (i.e., 90+ day). Why not use this definition, as this is the official WHO definition? I think the main results should be tested against this definition of post-covid for robustness. (Additionally, the strongest associations were found for "Other non-specific COVID-19 related disorders" (Fig. 1), which according to Appendix A seem like diagnoses partly describing the original COVID-19. This makes me wonder if these associations survive if starting to classify as post-covid at 90+ days.)

Thank you for this thoughtful suggestion. Regarding your first point about PCC definitions, as we described in the introduction, methods and discussion sections, the definition of PCC was inconsistent over time in the scientific literature. Moreover, definitions were derived using different study designs, time periods and populations which led to a large amount of uncertainty. We started follow-up at 30 days because we were attempting to remove health outcomes related to the acute stage of illness. At the time of writing the protocol, we conducted a literature review to inform our approach regarding the definition of 'acute illness' and the start of post-COVID conditions. Our results are included in **Table 1** of this response letter. Of the 11 large-scale studies we identified that described the association between prior vaccination and PCC, the majority (9 studies) used \$\leq 4\$ weeks to define

the acute stage of illness, and hence start follow-up after 30 days (T_0). Although there is no single agreed upon definition of PCC globally, this definition is consistent with the Centers for Disease Control definition of post-acute COVID-19 (<https://www.cdc.gov/coronavirus/2019-ncov/long-term-effects/index.html>). Therefore, for continuity and to allow comparisons with prior work, we decided to use the same definition of PCC as used in a prior study (Tartof *et al.*, 2022), which defined PCC as pre-specified conditions occurring 30 days – 6 months following diagnosis. However, we recognize that other sources describe post-acute COVID-19 as persistence of symptoms or development of sequelae after longer time periods such as 3-6 months (Greenhalgh *et al.*, 2020; Shah *et al.*, 2021), including WHO and NICE (UK). Therefore, to enhance the comparability of our findings with these sources, we have now included a sensitivity analysis whereby the follow up period is adjusted to 90 days- 6 months following the date of positive SARS-CoV-2 test (**Appendix K; Figure 2** below). We observed that overall the associations were stronger when follow up started later, reinforcing the findings overall that vaccination appears protective against the long-term sequelae of COVID-19. We have now included a brief summary of this sensitivity analysis, and the rationale for this sensitivity analysis in the limitations section (Lines 362-368).

Figure 2 (also **Appendix K** in revised manuscript). Association between prior vaccination and PCC when including proximal and distal time periods (30 days- 6 months) vs. distal periods (90 days – 6 months) of follow up

Regarding the second point you raised about “Other non-specific COVID-19 related outcomes”, this is an important point - thank you for identifying it. We decided to include “Other non-specific COVID-10 related disorders” as a separate outcome category because it could represent PCC outcomes affecting a range of body systems. It may also overrepresent symptoms consistent with the acute phase of COVID-19 illness, which may be more sensitive to changes in the follow-up time window. However, “Other non-specific COVID-19 related outcomes” remained strongly associated with vaccination status in the current study after restricting events to those occurring 90 days- 6 months (as opposed to 30 days – 6 months) in our sensitivity analysis (**Figure 2** in this response letter; **Appendix K** in the revised manuscript).

Table 1. Literature search results showing variety of PCC definitions used by authors of prior studies

Reference & Region	Study period	Study population	Design	Time cut-off for PCC	PCC Outcome definition
Marra et al., (2022) Multiple countries	Feb 2020 – Nov 2021	250,578 adults aged 18+	Meta-analysis	Multiple definitions	Multiple definitions
Antonelli (2021) UK	Dec 2020- Jul 2021	4,731 patients aged ≥18 years positive for SARS-CoV-2	Matched case-control	>28 days post-infection	Self-reported (digital App) Long duration symptoms of COVID-19 (≥28 days)
Taquet (2022) USA	Jan – Aug 2021	18,958 persons with confirmed COVID-19.	Nested case-control	0 days to 6 months post-infection	Any diagnosis: abnormal bleeding, anxiety/depression, chest/throat pain, cognitive symptoms, fatigue, headache, myalgia and other pain, and death
Al-Aly (2022) USA	Jan - Oct 2021	33,940 participants with break-through infection	Case-control	30 days to 6 months post-infection	Pre-specified outcomes (Al-Aly et al. 2021) from diagnoses, medications and laboratory results: CVD, coagulation, fatigue, GI disorders, kidney disorders, mental health disorders, metabolic disorders, musculoskeletal disorders, neurologic disorders, and pulmonary disorders.
Zisis (2022) USA	Sept 2020- Dec 2021	25 225 vaccinated patients diagnosed with COVID-19 and approx. 1.5M controls.	Case-control	≥4 weeks from initial infection.	New, continuing, or recurrent symptoms. Health conditions assessed using ICD-10 codes.
Arjun et al. (2022) India	Apr– Sept 2021	487 adults COVID-19 positive in a single hospital	Cohort study	≥4 weeks from the date of diagnosis.	Self-reported fatigue, cough, loss of taste and smell and cognitive dysfunction
Kuodi (2022) Israel	Mar 2020 – Jun 2021	951 COVID-19 positive adults	Nested cross-sectional study	≥4 weeks from date of test	Self-reported physical, mental, and psychosocial consequences of COVID-19 (fatigue, headache, weakness in arms and legs, persistent muscle pain, loss of concentration, hair loss, sleeping problems, dizziness, persistent cough, shortness of breath)
Senjam (2021) India	Jan –April 2021	773 Adults who tested positive for COVID-19	Cross-sectional	4-12 weeks (short term) and >12 weeks (long-term) symptoms	Self-reported Long COVID symptoms
Simon (2022) USA	Jan 2020 - May 2021	Persons of all ages with positive COVID-19 test (220,460 unvaccinated vs. 17,796 fully vaccinated)	Retrospective cohort	12 - 20 weeks after COVID-19 diagnosis	Chest pain, palpitations, altered mental state, anorexia, chills, fatigue, fever, malaise, loss of sense of smell, loss of sense of taste, nasal congestion, sore throat, abdominal pain, diarrhoea, digestive changes, nausea, vomiting, arthralgia, muscle weakness, general weakness, myalgia, headache, cough and dyspnoea.
Peghn et al. (2022) Italy	March 2020 – May 2021	479 patients with COVID-19 during March-May 2020 followed up at 6 and 12 months	Retrospective study	Signs or symptoms lasting longer than 12 weeks	Self-reported signs and symptoms developed during or after an infection consistent with COVID-19, continuing for more than 12 weeks, and not explained by an alternative diagnosis.
Herman (2022) Indonesia	2020 - Dec 2021	442 people with positive COVID-19 tests.	Matched case-control	2 and 4 weeks after infection	Olfactory dysfunction (anosmia or hyposmia).

Comment 2: Outcome definition – symptoms included and symptom grouping

A stronger argument must be given these diagnoses, and groupings of diagnoses, should be used to characterize post-covid. According to the WHO, the most common symptoms of post-covid are (i) fatigue, (ii) shortness of breath (or difficulty breathing), (iii) memory, concentration or sleep problems, (iv) persistent cough, (v) chest pain, (vi) trouble speaking, (vii) muscle aches, (viii) loss of smell or taste, (ix) depression or anxiety, and (x) fever. What does results look like if you use these as you PCC categories? Now fatigue, which together with respiratory problems are considered the main symptoms, are grouped under “symptoms” – why? And, why did fatigue (and some other complaints) only have a 30d look back period? A stronger argument should be made for being inconsistent across diagnoses in terms of lookback period.

Thank you for this important comment. We have included the individual sub-conditions for the WHO definition of PCC in the supplementary appendix (**Appendix I**), and we observe the following associations:

- i) Fatigue/fever/malaise: 0.92 (0.88-0.96)
- ii) Dyspnea/abnormal breathing: 0.93 (0.89-0.98)
- iii) Memory difficulty/amnesia and sleep disorders: 0.92 (0.82, 1.04) and 0.97 (0.92, 1.03), respectively
- iv) Cough: 0.96 (0.92-1.00)
- v, vi) Chest pain/Throat pain: 0.90 (0.86-0.94)
- vii) Muscle aches (body aches/myalgia): 0.93 (0.91-0.96)
- viii) Loss of smell or taste (ENT disorders): 0.91 (0.86-0.96)
- ix) Depression and anxiety: 1.10 (1.04-1.16) and 1.05 (1.01-1.09), respectively
- x) Fever/fatigue/malaise: 0.92 (0.88-0.96)

Therefore, while we feel it is important to recognize that the definition used in the current study differs slightly from other research in the field (now described in more detail in the limitations section, lines 362-369), we do not think that the overall finding will be altered if we were to use a different overall definition of PCC using the above sub-conditions. Indeed, as noted above, all associations for the WHO definition appeared to demonstrate similarly protective effects of prior vaccination, with RRs generally below 1 except for depression and anxiety. In fact, by encompassing more sub-conditions that have previously provided evidence of an association with SARS-CoV-2 infection, we feel that our choice of outcome is more comprehensive than the WHO definition. It was also developed in consultation with subject matter experts from CDC at the time of protocol development.

Thank you for your question about different look-back periods. As described in our methods section, we used outcome-specific lookback durations based on clinical judgement, since it is appropriate to use shorter lookback periods for acute outcomes, particularly for outcomes that may occur frequently such as acute respiratory outcomes. This approach is commonly applied to vaccine safety studies and other epidemiological studies assessing multiple health outcomes (Nelson et al., 2023, Xu et al., 2022, Hechter et al., 2019). We are confident that this is the optimal approach since some outcomes would have led to large numbers of exclusions if longer lookback periods were applied. However, our revised manuscript now includes a sensitivity analysis whereby all individuals with any documentation of each PCC outcome in the year prior to the index date are excluded from the analysis. As observed in **Figure 3** below and **Appendix J** in the revised manuscript, the associations were mostly unchanged, and hence we feel that applying a uniform look-back period to all outcomes would not have substantially altered the associations.

Figure 3. Original results (A) compared with sensitivity analysis excluding individuals with documentation of each PCC outcome in the year prior (B) – Also in **Appendix J** of the revised manuscript.

Comment 3: Statistical analyses – controlling for prior healthcare utilization

There are several studies suggesting that some cases of post-covid could have a considerable psychological mechanism to it, perhaps related to health anxiety. I think the analyses would be more robust if the authors also presented a version where they controlled for whether or not the individual in the previous year had a psychological diagnosis. Table 1 provides clear indications that there are significant differences between the groups in the year prior to infection. This is an indication that there still may be substantial confounding lurking in the background.

Thank you for this comment. We agree that any analysis describing PCC by vaccination status is susceptible to confounding by healthcare seeking behavior, which could be particularly true for mental health outcomes. Indeed, in exploratory analyses, we observed that mental health outcomes were more common among vaccinated individuals at baseline than unvaccinated individuals, despite matching on important demographic characteristics – more so than other conditions (results not shown). We suspect that this issue is consistent and ubiquitous across all studies that include mental health outcomes in their selected PCC definition. However, to investigate any potential impact of this bias on our main associations, we conducted a sensitivity analysis excluding all persons with a history of each PCC condition within one year prior to the analysis (Appendix J of revised manuscript; Figure 3 of this response letter). We did not observe a large difference in effect estimates, nor did we observe different changes in association between each PCC category after completing this sensitivity analysis. Hence, this indicates that any potential bias related to prior disease did not alter the associations, including mental health outcomes. We have included this language in our limitations section of the revised manuscript (Lines 351-354).

Comment 4:

Minor issues

- line 254: “substantial reduction”? For the most well-known PCC symptoms (Fig 1), the RRs are above 0.9. I think “substantial” feels like overselling. I suggest you tone this down. Same goes for “clear evidence” (line 369).

Thank you for this kind suggestion. We have softened the language in these two places (lines 269 and 396 in the revised manuscript) according to your suggestions.

- line 299: “the enhanced protective effect of COVID-19 vaccination against PCC among children and adolescents is a novel and important finding.” This also comes across as a bit overselling. It looked like (cf. Fig 2) there only were two (gastro and respiratory) out of 13 types of symptoms where there was a significant difference. Or did I misread your figure?

Thank you for this comment. We agree that perhaps the language was overselling this point. However, we still feel that due to the understudied nature of pediatric populations with respect to PCC, this is still an important finding. Therefore, we have softened the language in the revised manuscript (lines 313-314).

- Subgroup analysis: you chose age and virus type to be included in your main subgroup analysis – what was the rationale? A short motivation for these would be helpful.

We decided on our sub-group analyses a priori. We opted to include minimal subgroup analyses in the main text to keep the primary message of the analysis very clear. Also, we were cognizant of the potential for multiple sub-group analyses to increase the chances of identifying spurious associations. The specific reasons for including age and time period of infection subgroup analyses in the main text were as follows:

- i) Age is an important factor when describing PCC, with prior evidence suggesting that PCC differs greatly by age group (Taquet *et al.*, 2022).
- ii) Time period of infection, roughly approximating waves of infection (pre- and post-Omicron), since at the time of writing the manuscript, there was a lack of studies covering the time period during which Omicron was the dominant circulating variant.

- It would be helpful with a table showing the number of events for each of the diagnoses underlying Figure 1.

Thank you for this suggestion. We have included a column with the number of events for each of the diagnosis categories in Figure 1.

- In the unmatched analysis, the RR for mental disorders is among your strongest associations (fairly strong association

and tight Cis). This makes me a bit worried about confounding, although attenuated by the matching. How do you interpret this? Are there additional analysis you could do to investigate where this comes from? To speculate: it seems like those taking the vaccine have higher levels of mental health issues, and the vaccine could be one way to reduce health anxiety related to being infected by a fairly unknown, and potentially dangerous, virus.

Thank you for highlighting this to us. We agree, as mentioned in an earlier comment, that vaccination status is highly correlated with healthcare seeking behavior, as are some of our PCC outcomes (particularly mental health outcomes). Also, healthcare seeking behavior is difficult to characterize and hence challenging to control for. However, we have attempted to further control for the effects of baseline conditions – a close correlate of healthcare seeking behavior - on associations using an additional sensitivity analysis excluding individuals with each PCC documented in the year prior to their index date (**Appendix J**). However, results of the main analysis were mostly unchanged, indicating that the effect of any bias is not directional or specific to a particular PCC outcome.

- Line 66: I think you should be more specific in stating the aims of the study.

Thank you for this suggestion. The primary aim of our study was to investigate the overall association between prior vaccination and PCC. We clearly state this overall aim in a stand-alone paragraph at the end of the introduction (Lines 66-67): “This study aims to assess the association between prior COVID-19 vaccination status and incident PCC among individuals with SARS-CoV-2 infection enrolled at 8 large integrated healthcare systems.”

Reviewer 2

General comments

In this study, the authors sought to assess whether the incidence of post-COVID condition differed between those with post-vaccination SARS-CoV-2 infections and those without prior vaccination at the time of infection. This question has been addressed in several prior studies that are referenced in the introduction. The main advantage of the present study compared to existing studies is the increased sample size. However, it unfortunately comes with several major sources of bias (not present in previous studies) which cannot be addressed by increasing sample size (indeed increasing sample size only makes bias more apparent).

Thank you for your comment here. We agree that our study has many strengths which set it apart as a robust and unique analysis to address this important research question. The authors also understand your concerns around bias, some of which are ubiquitous in epidemiological research and hence unavoidable. However, we have tried our best to mitigate where these concerns could have potentially biased our results. Otherwise, if this was not possible, we have attempted to identify the direction of any potential bias and explain any consequences on the interpretation of findings in the limitations section.

Comment 1: Censoring for death without accounting for death as a competing risk: people who are vaccinated are less likely to die and thus more likely to develop any of the other outcomes, thus biasing all risks towards higher values.

Thank you for this thoughtful comment. We recognize that for time-to-event analysis, death may be a competing risk for some PCC outcomes and thus survivorship bias may be present. This means that for outcomes much rarer than death (which is the minority of outcomes investigated), risk estimates might be driven by death rather than the outcome of interest. However, unlike other prior studies (Maja *et al.*, 201, Taquet *et al.*, 2022), we believe that our analysis mitigates some of the impact of competing risks since our risk estimates are generated using a Poisson regression approach. However, there is still a possibility for survival bias if death rates are different between the exposure groups (vaccinated and unvaccinated). However, we performed a sensitivity analysis to assess this and we discovered that death occurred very rarely (<0.4%), and was similar between the two exposure groups (ASD = 0.023). This is likely due to matching on important determinants of death such as age and sex.

Comment 2: Including individuals in the cohort who can never develop the outcome (i.e. those with the outcome before

the exposure). The justification provided by the authors (to “ensure that the study population was consistent across all PCC events studied”) is unsatisfactory as it leads to biased estimates of risks.

Thank you for this comment. In the revised manuscript, we now include a sensitivity analysis excluding all individuals with a history of each PCC outcome (**Appendix J**). However, this approach introduces additional sources of bias and complicates the interpretation of associations by including different study populations for different PCC outcomes. Hence, the effect sizes are not completely comparable across all estimates and are heavily influenced by age. Another possible approach excluding all individuals with any history of PCC outcome would severely limit the representativeness and sample size since many outcomes are common, and are often skewed towards older age.

Comment 3: The use of “censoring” but without any time-to-event analysis. Risk ratios calculated after excluding censored individuals can be biased. To illustrate this point, imagine that Group A and Group B have 1000 people each at the beginning of a time window and both groups have 100 events during that time window, but fewer members of Group B are censored during that time window (e.g. 150 vs 50 in Group A). The risk ratio would be 1.12 $(=(100/850)/(100/950))$ even though both groups have the same number of events and same number of people at risk at the beginning of the time window.

Thank you for this comment. Individuals in the study were followed from 30 days to 6 months or until a censoring event occurred (e.g., termination of health plan membership, receipt of an additional dose of COVID-19 vaccine, re-infection, or death), whichever occurred first. We did not exclude censored individuals; instead, we excluded the time after the censoring event in the risk ratio calculation. For example, if an individual was disenrolled at the start of month 4, this individual only contributed person-time from 30 days to the end of month 3 for the analysis. We used a Poisson regression model to compare the PCC event rates between the vaccinated and unvaccinated groups, accounting for the length of follow-up. Therefore, the resulting estimate is unbiased. To use your example above, each group has 100 events if all individuals are followed for the full follow up period (i.e., with no censoring events). However, with the introduction of censoring, we would expect to observe fewer than 100 events in each group. Assuming events are evenly distributed across a time window, the event rate, calculated by dividing the number of events by the total person-time, remains the same in both groups, resulting in a risk ratio equal to 1.

Comment 4: The lack of any adjustment for individual comorbidities. Only adjustment for Charlson Comorbidity Score is used but individual comorbidities might still be unbalanced between cohorts, and different comorbidities might have different associations with the outcomes.

Thank you for your comment. We agree that unvaccinated and vaccinated populations likely differ with respect to many demographic, clinical, lifestyle and healthcare behavioral factors that are themselves related to comorbidities. However, by matching on age, sex, severity of initial infection, time of infection and race/ethnicity, we feel that any imbalance in these important confounding factors are minimized, including the existence of baseline disease. Indeed, when comparing the baseline comorbidity status for individual disorders between the two cohorts, we do not see a large difference in the proportions (see **Table 2** below), with all ASD <0.1.

Table 2. Baseline proportions with individual comorbidities between the vaccinated and unvaccinated populations

Comorbidity	Unvaccinated (N=161531)	Vaccinated (N=161531)	Total (N=323062)	Absolute Standardized difference
Myocardial infarction	1019 (0.6%)	1264 (0.8%)	2283 (0.7%)	0.02
Congestive heart failure	1683 (1.0%)	2202 (1.4%)	3885 (1.2%)	0.03
Peripheral vascular disease	4909 (3.0%)	6079 (3.8%)	10988 (3.4%)	0.04
Cerebrovascular disease	1396 (0.9%)	1618 (1%)	3014 (0.9%)	0.01
Dementia	703 (0.4%)	781 (0.5%)	1484 (0.5%)	0.01
Chronic pulmonary disease	14662 (9.1%)	15857 (9.8%)	30519 (9.4%)	0.03
Connective tissue disease-rheumatic disease	1218 (0.8%)	1534 (0.9%)	2752 (0.9%)	0.02
Peptic ulcer disease	223 (0.1%)	312 (0.2%)	535 (0.2%)	0.01
Diabetes with complications	3902 (2.4%)	5934 (3.7%)	9836 (3.0%)	0.07
Diabetes without complications	6149 (3.8%)	8121 (5%)	14270 (4.4%)	0.06

Paraplegia and hemiplegia	319 (0.2%)	378 (0.2%)	697 (0.2%)	0.01
Renal disease	3125 (1.9%)	4476 (2.8%)	7601 (2.4%)	0.06
Moderate or severe liver disease	162 (0.1%)	260 (0.2%)	422 (0.1%)	0.02
Metastatic carcinoma	532 (0.3%)	624 (0.4%)	1156 (0.4%)	0.01
AIDS	53 (0.0%)	174 (0.1%)	227 (0.1%)	0.03
Mild liver disease	2949 (1.8%)	3541 (2.2%)	6490 (2.0%)	0.03
Cancer except metastatic carcinoma	1755 (1.1%)	2117 (1.3%)	3872 (1.2%)	0.02
Number of Charlson Comorbidities				0.09
0	130848 (81.0%)	126015 (78.0%)	256863 (79.5%)	
1 to 2	24058 (14.9%)	26396 (16.3%)	50454 (15.6%)	
3 & more	6625 (4.1%)	9120 (5.6%)	15745 (4.9%)	

Comment 5: The lack of matching for important covariates with influences on outcomes that might be greater than the influence of age or sex, such as prior influenza vaccination which is reported to be 30.1% vs. 66.7% in the two cohorts, and likely captures vaccine hesitancy.

The increased statistical power offered by the larger sample size is also underutilized. The authors report that they are able to examine rarer outcomes, such as “acute respiratory disorders, cardiac arrhythmias/postural orthostatic tachycardia syndrome (POTS), neurological disorders, and sensory disorders” but most of these outcomes are already investigated in previous studies including reference [15] and [17].

Thank you for pointing this out. We agree that the large sample size represents a particular strength of our study. However, despite this, we did not place a large emphasis on the results of rarer sub-conditions such as POTS for several reasons:

- 1) Space limitations precluded the ability to discuss each of the 51 PCC sub-conditions separately in the manuscript, particularly comparisons with prior literature.
- 2) We were cognizant of the impact of multiple outcome assessments on the potential to identify spurious associations, particularly since we conducted multiple subgroup analyses.
- 3) Despite our large sample size, some of the 51 PCC sub-conditions still occur rarely in this study population. This is demonstrated by the wide confidence intervals in **Appendix I** showing the lack of statistical power for most PCC sub-conditions.

Therefore, given the above constraints, we have opted to present the results of all 51 PCC outcomes in the Appendix to show their directional concordance (despite the lack of statistical power), in the event that readers are interested in a particular sub-condition, and would like to compare the associations with prior work. Since no prior study has had the power to present all 51 conditions in a single study, we still feel that the ability to assess individual sub-conditions is a particular strength of the current study. Therefore, we have opted to keep the wording as is.

Comment 6: Finally, to demonstrate the coverage of the dataset, it would be informative to report the overall vaccination coverage in the whole database (i.e. including in those who did not have COVID-19) and compare this to vaccination coverage in the American population during the same period.

Thank you for this comment. Assessing the coverage of the entire population of patients over the study period is beyond the scope of our study since the analysis does not include individual-level data for all members of the 8 integrated healthcare systems included in this study.

It is difficult to directly compare vaccination coverage within the VSD population with the overall US vaccination coverage due to the potential lack of representativeness and a lack of accurate national coverage estimates, as others have expressed previously (Razzaghi et al., 2021, Xu et al., 2021). Indeed, the study population had continuous healthcare insurance coverage, and hence it is likely that vaccination coverage will be slightly higher in the current study than among the general US population. However, this difference is likely minimal since the population is thought to be generally representative of the US population with regard to several demographic and socioeconomic characteristics (Sukumaran et al., 2015). This is described in the limitations section (lines 389-392).

Reviewer 3

General comments

Overall – well conducted study which makes an important contribution to the field with a robust study design and large sample size. Specifically, the study supports previous research regarding the positive effects of vaccination on persistent symptoms following SARS-CoV-2 infection and goes beyond existing studies by exploring this relationship in children and adolescents.

Specific comments

Comment 1: The authors use PCC but other terms are in use, including Long COVID, post-covid syndrome. Please briefly describe the differences and comment as to why PCC is used in this study.

Thank you for this comment. As discussed above, the definition and terminology of “Long COVID” changed over time, and we do not feel there is sufficiently clear rationale available in the literature to explain the use of each term. For the current manuscript, we settled on ‘Post-COVID conditions (PCC)’ as the agreed upon terminology at the time of writing following discussions with the CDC’s Post-COVID Conditions working group. This terminology has also been adopted by WHO (<https://www.who.int/teams/health-care-readiness/post-covid-19-condition>) and CDC (<https://www.cdc.gov/coronavirus/2019-ncov/long-term-effects/index.html>).

Comment 2: Please add a bit more context in terms of when and how vaccination was rolled out in US or in the geographical area covered by VSD.

Thank you for this suggestion. Person of all ages with a positive SARS-CoV-2 test between March 1st 2021 – February 28th 2022 were included in this analysis. During this period, the majority of infections occurred during the Omicron wave and vaccination coverage across the US was high (<https://covid.cdc.gov/covid-data-tracker/#vaccination-trends>). Hence, there were more vaccinated than unvaccinated persons in the unmatched study population (see **Appendix C** of the manuscript). We have included this information in the results section: “Most patients with SARS-CoV-2 were identified in December 2021 and January 2022, approximately reflecting the time period during which Omicron was the dominant circulating variant and COVID-19 vaccination coverage was high in the US (**Appendix C**)” (Lines 192-194).

Comment 3: Also please briefly explain the SARS-CoV-2 testing policy (free, universal, mandatory, etc....) during the study period as this information would give the reader an indication of the likelihood of missed infections through lack of testing. Could testing behaviour or test availability/policy influence presence or absence of a SARS-CoV-2 test result in EHR? How is this likely to affect the findings – please briefly discuss in the section of the discussion on limitations.

Thank you for bringing up this important point about testing practices. Although testing practices may have varied slightly between VSD sites due to regional supply or internal policies, the study period covers a time period when SARS-CoV-2 tests were generally available in the US. If general population-level testing behaviors were present in this population, we feel that these would have impacted vaccinated and unvaccinated individuals similarly and hence would not have biased the results. We have included the following sentence in the limitations section: “SARS-CoV-2 tests were widely available during the study period and therefore testing behaviors should not have differed by severity of illness or vaccination status” (Lines 347-349).

Comment 4: It would be helpful to provide greater context from other studies which have done similar analyses (principally those mentioned). The discussion refers to these studies but further consider the similarities and differences between the presented work and existing analyses.

Thank you for this comment. We agree that it is important to compare our results against the findings of similar studies.

In our study, instead of making direct comparisons with specific studies between specific PCC outcomes, we compare the overall findings, and summarize the overall shortfalls of prior studies. We also highlight where our study addresses these, particularly with regard to sample size, the availability of standardized and complete datasets, and the robust statistical methodology of our study. We opted not to compare results of prior work for each individual 13 PCC outcome, for two main reasons:

- 1) As mentioned in the introduction (lines 59-62) and the discussion (lines 278-283), previous studies vary greatly in their study designs, time periods, PCC definitions and study populations. This makes it challenging to compare associations for each PCC category across studies.
- 2) As in other studies, our results cover a wide range of PCC outcomes. Therefore, we are not able to compare each outcome with all prior studies due to space limitations. For this reason, we opted to focus on the overall finding and results of the pre-specified subgroup analysis rather than specific PCC outcomes or sub-conditions.

Comment 5: Regarding the VSD and eight integrated healthcare systems. Can you give more detail in terms of geographical area covered and sociodemographic profile of underlying population and any information regarding quality/completeness of data in VSD or validation studies on this dataset.

Thank you for this suggestion. The VSD population is comprised largely of individuals with health insurance coverage, and the VSD findings may not be generalizable to uninsured populations. We have discussed this potential limitation in our limitations section (lines 389-390). However, although the extremes of income distribution may be under-represented, the VSD population has been found to have similar demographic characteristics to the catchment areas served by the VSD health plans (Lines 390-392). With the increase in external vaccine administrations during the national vaccination rollout in the US, a recent study was conducted to assess the ability of the eight VSD sites included in this manuscript to receive and integrate COVID-19 vaccine data from external settings (Groom et al., 2022). The results of this study demonstrated the timely integration of Immunization Information Systems (IIS) data from non-traditional and external settings into patient EHR (now described in lines 75-76 of revised manuscript).

Comment 6: Regarding sentence “Therefore, multiple positive tests per patient were included if they occurred at least 90 days apart.” Could you add a reference to justify this assumption. Consider adding, for example reference “UK Government. Investigation and management of suspected SARS-CoV-2 reinfections: a guide for clinicians and infection specialists 2021. Available at: <https://www.gov.uk/government/publications/covid-19-investigation-and-management-of-suspected-sars-cov-2-reinfections/investigation-and-management-of-suspected-sars-cov-2-reinfections-a-guide-for-clinicians-and-infection-specialists>.”

Thank you for this helpful comment. We agree that this reference is helpful as a rationale for the 90-day definition of reinfection used in the current study. We have included this reference in the suggested place (line 85).

Comment 7: Patients who received either a Janssen COVID-19 vaccine or a COVID-19 vaccine not routinely administered in the US were not included in the analysis. Were there sufficient numbers to do a sub analysis by type of vaccine? Might be of interest for countries using other vaccine types.

Thank you for this comment. We made the decision to exclude Janssen COVID-19 vaccine recipients because this population was very small (comprising around 3% of vaccinated cases across the cohort) and may have systematically differed from the rest of the study population by several factors that could bias the risk estimates. First, during the study period (on April 13–23, 2021), CDC and FDA recommended a pause in use of Janssen vaccine after they received reports of six cases of cerebral venous sinus thrombosis with thrombocytopenia among Janssen vaccine recipients (Shay *et al.*, 2021). Hence, persons who may have previously been eligible for the Janssen vaccine no longer had the opportunity to be vaccinated after this period, and therefore a sub-group analysis of this study population would be difficult to compare against other vaccinated individuals. Second, persons who opted to receive Janssen may have differed systematically with respect to clinical risk factors for COVID-19 illness (and therefore perhaps PCC), since the single dose was seen as a ‘rapid immunity’ option for persons who may have been particularly vulnerable to severe illness or who had multiple exposures to SARS-CoV-2. Therefore, whilst this may have been of interest to other countries, we hesitate to complete this subgroup analysis due to the risk of producing a biased estimate with limited statistical power. We included this additional rationale in lines 89-93 of the methods section: “if patients received either a Janssen (Johnson & Johnson) COVID-19 vaccine or a COVID-19 vaccination not routinely administered in the US, they were not included in the analysis since it is likely that these patients systematically differ from mRNA vaccine recipients and the limited sample size would preclude meaningful subgroup analysis among these individuals”.

Comment 8: WHO and NICE in UK, for example, use different definitions, for example Long COVID, post covid syndrome, persistent symptoms for ≥ 12 weeks. As a sensitivity analysis (and perhaps to be included as supplementary material) could the main analysis be replicated using at least one other alternative definition to see if the findings are consistent.

Thank you for this thoughtful suggestion. Regarding your first point about PCC definitions, as we described in the introduction, methods and discussion sections, the definition of PCC in scientific literature was inconsistent over time. Moreover, these definitions were derived using different study designs, time periods and populations. We started follow-up at 30 days because we were attempting to remove health outcomes related to the acute stage of illness. At the time of writing the protocol, we conducted a literature review to inform our approach regarding the definition of ‘acute illness’ and the start of post-COVID conditions. Our results are included in **Table 1** of this response letter. Of the 11 large-scale studies we identified that described the association between prior vaccination and PCC, the majority (9 studies) used 4 weeks following infection as the start of follow-up (T_0). Of note, it is recognized that there is no single agreed upon definition of PCC globally. Therefore, to allow comparisons with prior work, we decided to use the same definition of PCC as used in a prior study (Tartof *et al.*, 2022), which was 30 days following diagnosis. Our definition is also consistent with the Centers for Disease Control definition of post-acute COVID-19, defined as emergent, recurring or persistent symptoms occurring ≥ 4 weeks following acute infection with COVID-19 (<https://www.cdc.gov/coronavirus/2019-ncov/long-term-effects/index.html>). However, we recognize that other sources describe post-acute COVID-19 as persistence of symptoms or development of sequelae after longer periods from the onset of acute symptoms of COVID-19 (Greenhalgh *et al.*, 2020; Shah *et al.*, 2021), including WHO and NICE. Therefore, to enhance the comparability of our findings, we have now included a sensitivity analysis whereby follow up starts at 90 days (3 months) following the date of positive SARS-CoV-2 test (**Appendix K**). We observed that the associations appeared stronger when follow up started later, reinforcing our primary finding that vaccination appears protective against long-term sequelae of COVID-19. We have included a brief summary of this issue and the results of our sensitivity analysis in the limitations section of the revised manuscript (lines 362-368).

Comment 9: Regarding “weighted Charlson comorbidity score” - Could you provide a reference for this or more detail of this score in the text or supplementary material, including how it is calculated and its interpretation.

The Charlson Comorbidity index consists of pre-selected 17 chronic conditions as defined by ICD-10 codes (see **Table 3** below; Quan *et al.*, 2005) in 12 months prior to the date of positive SARS-CoV-2 test. This index is utilized as a weighted score, with each condition being assigned a standardized weight (see **Table 4** below; Charlson *et al.*, 1987). The comorbidity which has a higher risk of mortality is assigned a higher weight. We have now included a reference in the manuscript that describes the weighted Charlson Comorbidity index calculation (Line 149).

Table 3. ICD-9-CM and ICD-10 coding algorithms for Charlson Comorbidities

[REDACTED]

Table 4. Weights assigned to conditions included in the weighted Charlson Comorbidity Index score

[REDACTED]

Comment 10: From your analysis can you say anything specific about "brain fog" or memory or cognitive symptoms which have been widely reported in those with persistent symptoms following covid?

As outlined in **Appendix A**, ICD10 codes related to Amnesia/memory difficulty were included in our PCC definition (R41*). The ICD10 code often used to document 'brain fog' in the literature is "**R41.9**: Unspecified symptoms and signs involving cognitive functions and awareness". We have included this particular ICD-10 code in our PCC definition alongside other outcomes related to memory and cognitive function. For a more specific effect size, please refer to **Appendix I**, which shows that amnesia and memory problems were associated with a similar effect size as compared with most other neurological or nervous system disorders (RR: 0.92, 95% CI: 0.82-1.04). However, with this level of granularity, there was a lack of events and hence the confidence intervals are wide. We recognize that "brain fog" may be of particular interest to specific audiences, but given that a total of 51 PCC outcomes were included in this analyses, we lean towards not selecting individual sub-conditions to discuss in the manuscript.

Comment 11: Comment on potential (if any) misclassification of infection status, as per comments above regarding quality/completeness of dataset (VSD) and testing practices.

It is possible that persons who received a confirmatory positive SARS-CoV-2 test (and hence were eligible for this study) were systematically different than persons infected with SARS-CoV-2 who did not receive a confirmatory test during the study period. However, we feel that the impact on the associations presented in our study would be minimal since this would likely have impacted vaccinated and unvaccinated individuals similarly. Importantly, we match on date of test, so any general changes to population-level testing behavior will have been controlled for in the current analysis.

Comment 12: The conclusions appear overall well supported by the work, but it would be interesting to understand better the relationship between race/ethnicity and profile of EHR population (compared to underlying overall population) to disentangle what is a bias in EHR vs what is related to COVID-19 vaccination and PCC.

Since the VSD population is comprised of individuals with health insurance coverage, the VSD findings may not be generalizable to uninsured populations, particularly for those variables which are closely correlated with socio-demographic characteristics, such as race/ethnicity. We have been upfront about this potential limitation in our limitations

section (lines 389-392). However, although the extremes of income distribution may be under-represented, the VSD population has been found to have similar demographic characteristics to the catchment areas served by the VSD health plans, including race/ethnicity (Sukumaran et al., 2015; Koebnick et al., 2012). Importantly, we adjusted for race/ethnicity in our multivariable models, since it is a potentially important confounder underlying the association between vaccination status and PCC. Therefore, any associations that were identified should not be impacted by differences in this variable. We did not feel that race/ethnicity should be considered for a sub-group analysis since populations of different race/ethnicity differ with respect to other characteristics which are difficult to account for, such as healthcare seeking behavior. Also, the manuscript currently includes multiple sub-group analyses and we are cognizant of the impact on multiple testing.

Comment 13: Table 1 - Please give exact p values throughout.

We reported exact p values with 3 decimal places in **Table 1** (i.e., the only place where p values are presented). Due to the large sample size, most baseline covariates except matching variables had a p value less than 0.001.

Comment 14: Figures 1-3 - Figures 1,2,3 the category "symptoms" what organ/system does this refer to? Or is it non-specific symptoms - please clarify in text or in a footnote.

Thank you for this comment. We are cognizant of the fact that the PCC category 'symptoms' is perhaps ambiguous regarding the conditions included. We opted to include this PCC category since these 'symptoms' were thought to represent conditions that are etiologically similar to one another such as viral infections, and hence may display systematically different associations with vaccination status compared with other PCC categories that were categorized based on specific body systems. However, to address this, each figure and table included in the manuscript includes a separate footnote: "PCC category 'symptoms' included headache, body ache/myalgia, fever/malaise/fatigue, lymphadenopathy, weight loss, or vertigo." We also include this description when symptoms are first mentioned in the results section (Lines 223-224).

In addition to the above comments, we have corrected spelling and grammatical errors. We look forward to hearing from you in due time regarding our submission and to respond to any further questions and comments you may have.

Yours Sincerely,

Debbie E. Malden, DPhil, MSc

(Corresponding author)

Epidemiologist

Kaiser Permanente Southern California

Department of Research & Evaluation

References

- Charlson ME, Pompei P, Ales KL, MacKenzie CR. A new method of classifying prognostic comorbidity in longitudinal studies: development and validation. *J Chronic Dis.* 1987;40(5):373-83. doi: 10.1016/0021-9681(87)90171-8.
- Greenhalgh T, Knight M, A'Court C, Buxton M, Husain L. Management of post-acute covid-19 in primary care. *BMJ.* 2020;370: m3026. doi: 10.1136/bmj.m3026
- Groom HC, Crane B, Naleway AL, Weintraub E, Daley MF, Wain K, Beth Kurilo M, Burganowski R, DeSilva MB, Donahue JG, Glenn SC, Goddard K, Jackson ML, Kharbanda EO, Lewis N, Lou Y, Lugg M, Scotty E, Sy LS, Williams JTB, Irving SA. Monitoring vaccine safety using the vaccine safety Datalink: Assessing capacity to integrate data from Immunization Information systems. *Vaccine.* 2022 Jan 31;40(5):752-756. doi: 10.1016/j.vaccine.2021.12.048. Epub 2021 Dec 31.
- Hechter RC, Qian L, Tartof SY, Sy LS, Klein NP, Weintraub E, Mercado C, Naleway A, McLean HQ, Jacobsen SJ. Vaccine safety in HIV-infected adults within the Vaccine Safety Datalink Project. *Vaccine.* 2019 May 31;37(25):3296-3302. doi: 10.1016/j.vaccine.2019.04.080. Epub 2019 May 4. PMID: 31064675; PMCID: PMC6538462.
- Koebnick C, Langer-Gould AM, Gould MK, Chao CR, Iyer RL, Smith N, et al. Sociodemographic characteristics of members of a large, integrated health care system: comparison with US Census Bureau data. *Perm J* 2012;16: 37–41.
- Nelson JC, Ulloa-Pérez E, Yu O, Cook AJ, Jackson ML, Belongia EA, Daley MF, Harpaz R, Kharbanda EO, Klein NP, Naleway AL, Tseng HF, Weintraub ES, Duffy J, Yih WK, Jackson LA. Active Postlicensure Safety Surveillance for Recombinant Zoster Vaccine Using Electronic Health Record Data. *Am J Epidemiol.* 2023 Feb 1;192(2):205-216. doi: 10.1093/aje/kwac170.
- Quan H, Sundararajan V, Halfon P, Fong A, Burnand B, Luthi JC, Saunders LD, Beck CA, Feasby TE, Ghali WA. Coding algorithms for defining comorbidities in ICD-9-CM and ICD-10 administrative data. *Med Care.* 2005 Nov;43(11):1130-9. doi: 10.1097/01.mlr.0000182534.19832.83.
- Razzaghi H, Meghani M, Pingali C, Crane B, Naleway A, Weintraub E, Kenigsberg TA, Lamias MJ, Irving SA, Kauffman TL, Vesco KK, Daley MF, DeSilva M, Donahue J, Getahun D, Glenn S, Hambidge SJ, Jackson L, Lipkind HS, Nelson J, Zerbo O, Oduyebo T, Singleton JA, Patel SA. COVID-19 Vaccination Coverage Among Pregnant Women During Pregnancy - Eight Integrated Health Care Organizations, United States, December 14, 2020-May 8, 2021. *MMWR Morb Mortal Wkly Rep.* 2021 Jun 18;70(24):895-899.
- Shah W, Hillman T, Playford ED, Hishmeh L. Managing the long term effects of covid-19: summary of NICE, SIGN, and RCGP rapid guideline. *BMJ.* 2021;372: n136. doi: 10.1136/bmj.n136
- Shay DK, Gee J, Su JR, et al. Safety Monitoring of the Janssen (Johnson & Johnson) COVID-19 Vaccine — United States, March–April 2021. *MMWR Morb Mortal Wkly Rep* 2021;70:680–684. DOI: <http://dx.doi.org/10.15585/mmwr.mm7018e2>
- Sukumaran L, McCarthy NL, Li R, et al. Demographic characteristics of members of the Vaccine Safety Datalink (VSD): a comparison with the United States population. *Vaccine* 2015;33:4446–50.
- Tartof SY, Malden DE, Liu IA, Sy LS, Lewin BJ, Williams JTB, Hambidge SJ, Alpern JD, Daley MF, Nelson JC, McClure D, Zerbo O, Henninger ML, Fuller C, Weintraub E, Saydah S, Qian L. Health Care Utilization in the 6 Months Following SARS-CoV-2 Infection. *JAMA Netw Open.* 2022 Aug 1;5(8):e2225657.
- Xu S, Hong V, Sy LS, Glenn SC, Ryan DS, Morrisette KL, Nelson JC, Hambidge SJ, Crane B, Zerbo O, DeSilva MB, Glanz JM, Donahue JG, Liles E, Duffy J, Qian L. Changes in incidence rates of outcomes of interest in vaccine safety studies during the COVID-19 pandemic. *Vaccine.* 2022 May 20;40(23):3150-3158. doi: 10.1016/j.vaccine.2022.04.037. Epub 2022 Apr 18. PMID: 35465977; PMCID: PMC9013605.
- Xu S, Huang R, Sy LS, Glenn SC, Ryan DS, Morrisette K, Shay DK, Vazquez-Benitez G, Glanz JM, Klein NP, McClure D, Liles EG, Weintraub ES, Tseng HF, Qian L. COVID-19 Vaccination and Non-COVID-19 Mortality Risk - Seven Integrated Health Care Organizations, United States, December 14, 2020-July 31, 2021. *MMWR Morb Mortal Wkly Rep.* 2021 Oct 29;70(43):1520-1524. doi: 10.15585/mmwr.mm7043e2.

REVIEWER COMMENTS

Reviewer #1 (Remarks to the Author):

Thank you for the opportunity to review this paper again. The authors have done several adjustments to the manuscript to address the issues raised. Below I briefly revisit my comments from the previous round with my comments to the response (in italic):

-Comment 1: Outcome definition - time window

Thank you for a thorough response. I have no further comments related to this.

Comment 2: Outcome definition – symptoms included and symptom grouping

This is the critical point, in my opinion.

Thank you for calculating results using the WHO symptom list. However I could not find these results referred to in the revised manuscript, or in Appendix I in the version of the Supplementary Materials that I had, unless I looked at the wrong files.

I also register that these results (cf. RRs reported in the rebuttal letter) were much less strong when using this symptom grouping. Assuming you used Bonferri correction, «marginally protective» is my reading.

And related to these results: the vaccine has a marginally protective effect on the main/most prevalent symptoms of PCC. Why then, or what biological mechanism, can explain strong results for much more «exotic» symptoms, while almost no effect on the by far more prevalent symptoms? I think this needs to be explained, if the authors want to stick to this grouping.

In summary, I am still not convinced by your reasons for choosing a different grouping PCC conditions. Why deviate from the findings in the literature regarding the most prevalent symptoms? A much better rationale must be given. The rationale given in the paper is «We derived our PCC definition based on a CDC definition developed in 2021 used in an earlier analysis of VSD data [21].» This earlier study is JAMA Network Open study. Although surely a good study, I do not see why this papers should deviate from the symptom groupings used by more authoritative sources, including the WHO or NICE. For example, NICE has done a review of several studies of prevalent symptoms and ends up with a set of symptoms that closely mirrors WHO (see page 12 here, <https://www.nice.org.uk/guidance/ng188/evidence/evidence-review-b-prevalence-pdf-13307838590>). Neither do I see that your list corresponds to the symptom list suggested by the CDC (<https://www.cdc.gov/coronavirus/2019-ncov/long-term-effects/index.html>).

Comment 3 and minor comments:

No further comments. Thank you.

Reviewer #2 (Remarks to the Author):

I thank the authors for addressing some of my comments. The fact that individual comorbidities are well-balanced between groups is particularly reassuring.

However, I continue to have three concerns:

1) Poisson regression is unfortunately not less sensitive to competing risks than survival analyses. The risk of death might appear low at the population level, but it can affect different subgroups differently (e.g., pre-omicron, older people, etc.) and might therefore bias subgroup analyses. It is also possible that death is under-recorded, unless the data were linked to mortality registries.

2) My original comment on matching for prior influenza vaccination (an important determinant of whether one will receive a COVID-19 vaccine, as mentioned by the authors) seems to have been ignored. Linearly adjusting for such an important covariate is not satisfactory, and I would urge the authors to leverage the high sample size that they have to more tightly control for this important covariate.

3) The introduction is clear about how the authors compare their study to references [15] and [17], but less so on how they compare it to references [12] and [16], which both (especially [12]) have enough power to detect associations of interest and both have a 6-month follow-up. I strongly believe that replication of the findings of these studies in this separate and larger sample is worthy of publication, as it is an important research question. However, it should then be made more explicit in the introduction that this is the aim of the study, as I fail to see the novelty otherwise. Of note, reference [14] is unrelated to vaccination and should probably be removed from this passage in the introduction.

Reviewer #2 (Remarks on code availability):

The code is not available at that URL, only via direct email to the study authors.

Reviewer #3 (Remarks to the Author):

I would like to thank the authors for addressing my initial comments. The authors have provided a nicely detailed and thorough response to the comments from the previous review and have addressed my major concerns. Following the revision to the article, the authors have sufficiently improved their paper, in reaction to the comments I made.

March 2024

Ref: Response to reviewer's comments round II (NCOMMS-23-49571-T)

Dear Sir/Madam,

Thank you for your time and consideration reviewing our manuscript titled 'Post-COVID conditions following COVID-19 vaccination: A matched analysis of patients with SARS-CoV-2 from 8 large integrated healthcare systems' submitted to *Nature Communications*. On behalf of all coauthors, we appreciate your insightful comments on this manuscript, and we have altered our manuscript to reflect these suggestions.

Where appropriate, we have highlighted the changes made in response to your second round of comments in the enclosed revised version of our manuscript. In addition, please find a point-by-point response to specific comments and concerns outlined below in blue font.

Reviewer 1

Thank you for the opportunity to review this paper again. The authors have done several adjustments to the manuscript to address the issues raised. Below I briefly revisit my comments from the previous round with my comments to the response (in italic):

Comment 1: Outcome definition - time window

Thank you for a thorough response. I have no further comments related to this.

Comment 2: Outcome definition – symptoms included and symptom grouping

This is the critical point, in my opinion.

Thank you for calculating results using the WHO symptom list. However I could not find these results referred to in the revised manuscript, or in Appendix I in the version of the Supplementary Materials that I had, unless I looked at the wrong files.

I also register that these results (cf. RRs reported in the rebuttal letter) were much less strong when using this symptom grouping. Assuming you used Bonferri correction, «marginally protective» is my reading.

And related to these results: the vaccine has a marginally protective effect on the main/most prevalent symptoms of PCC. Why then, or what biological mechanism, can explain strong results for much more «exotic» symptoms, while almost no effect on the by far more prevalent symptoms? I think this needs to be explained, if the authors want to stick to this grouping.

In summary, I am still not convinced by your reasons for choosing a different grouping PCC conditions. Why deviate from the findings in the literature regarding the most prevalent symptoms? A much better rationale must be given. The rationale given in the paper is «We derived our PCC definition based on a CDC definition developed in 2021 used in an earlier analysis of VSD data [21].» This earlier study is JAMA Network Open study. Although surely a good study, I do not see why this papers should deviate from the symptom groupings used by more authoritative sources, including the WHO or NICE. For example, NICE has done a review of several studies of prevalent symptoms and ends up with a set of symptoms that closely mirrors WHO (see page 12 here, <https://www.nice.org.uk/guidance/ng188/evidence/evidence-review-b-prevalence-pdf-13307838590>). Neither do I see that your list corresponds to the symptom list suggested by the CDC (<https://www.cdc.gov/coronavirus/2019-ncov/long-term-effects/index.html>).

Response: Thank you for your follow-up comment to our original response regarding the outcome definition used in our paper.

We agree that this is an important topic. We are still learning a great deal about Post-COVID Conditions (PCC) and the definition is far from settled in the literature and across many organizations, including WHO, NICE and CDC. We are using a definition of Post-COVID Conditions that corresponds to the terms and definitions used by the U.S. Department Health and Human Services ([https://www.covid.gov/be-informed/longcovid/about*term__!w!!BZ50a36bapWJ!sjt4FbmTaquChezjZGT304xcZH6iOwPvux4jzdASgY4sF8ZGcHb9R_DXq9Z0WR3YLvtLt2yUDuLkC4\\$](https://www.covid.gov/be-informed/longcovid/about*term__!w!!BZ50a36bapWJ!sjt4FbmTaquChezjZGT304xcZH6iOwPvux4jzdASgY4sF8ZGcHb9R_DXq9Z0WR3YLvtLt2yUDuLkC4$)). CDC has consistently used this definition (or slight variations of this definition) in a number of publications, presentations and government reports (Hernandez-Romieu et al., 2022, Wanga et al., 2021, Hernandez-Romieu, 2020). As mentioned in the introduction, we have selected a broad definition for Post-COVID Conditions (i.e., this is a more sensitive than specific approach to capturing PCC). These conditions include all conditions and symptoms that we have found to occur more frequently among individuals following SARS-CoV-2 infection compared to non-SARS-CoV-2 infected individuals. Importantly, our definition also recognizes that individuals may experience an increased risk of system specific conditions such as diabetes, neurological disease, cardiovascular disease, or kidney disease. Symptoms included in the WHO and NICE definitions are narrower and focused more on symptoms that are more closely related to the acute stage of illness and unlikely to be explained by other causes. Furthermore, we feel that evidence for these PCC definitions developed by NICE and WHO are subject to important caveats which may explain some of the uncertainty they mention in each of the links you have provided.

For example, the list of symptoms described by NICE in the report that you shared (Table #1 below is an extract from page 12 of this report), was generated using evidence from 13 studies of non-hospitalized participants. We also cite some of these papers, and the symptoms covered in these studies cover more than those included in the NICE definition. Furthermore, we feel that these studies were subject to a number of important limitations, which may explain the wide range in prevalence estimates they cite in the table. Some of the uncertainty is likely attributed to the included studies reliance on self-reported survey data. As we discuss in the introduction section and discussion of our manuscript, self-reported data has a number of important caveats, including recall bias and a ‘healthy volunteer’ effect leading to a lack of representativeness. Therefore, while we recognize that NICE has attempted to standardize the definition of PCC in this report, we have opted to use a more comprehensive list of PCC outcomes developed by CDC based on electronic health care documentation that has been validated in the same study population at Kaiser Permanente Southern California previously.

Table 1. Extract from NICE report on literature review assessing symptoms of post COVID-19 conditions (PCC)

Symptom	Number of studies	Number of people (n)	Prevalence (range, %)
Loss of smell	8	3110	7% to 51%
Loss of taste	7	2960	5% to 51%
Shortness of breath	6	2999	8% to 71%
Chest pain	6	2999	6.9% to 44%
Joint pain	6	2999	2% to 31%
Headache	5	2849	5% to 38%
Fatigue	4	2823	27% to 87%
Palpitations	4	2510	10% to 32%
Fever	4	2710	2% to 11%
Cognitive impairment	2	679	2% to 29%

Furthermore, although very similar to the NICE definition, the WHO definition is vague regarding the origin of evidence for the included list of symptoms. Additionally, in the WHO website link that you provided, these symptoms are framed as “the most common symptoms” of post COVID-19 conditions, indicating that the complete post COVID-19 conditions list is more extensive than these 10 most common symptoms alone.

Despite these considerations, we agree that the post COVID-19 conditions defined by NICE and WHO are of public health importance and may be of interest to selected readers. Hence, whereas the original revisions did not include this list in the main analysis or the appendix as a stand-alone analysis (mostly for the reasons stated above), we have opted to include

these symptoms in a sub-analysis which is now included in the appendix of the revised manuscript (**Appendix L**). We also describe these results in the discussion (lines 365-369). However, please note that these outcomes were based on ICD10 diagnosis codes documented during healthcare encounters which may have led to incomplete capture of events for some symptoms such as “headache”, for example, which are less likely than other symptoms to result in healthcare utilization. Therefore, in comparison to the main analyses, statistical power is reduced for these outcomes. This could also be the reason underlying the marginally protective effect observed for the “main/most prevalent” symptoms of PCC since some non-specific outcomes such as symptoms of acute infections are more difficult to identify using ICD10 diagnoses. Since this measurement error is unlikely to differ systemically between vaccinated and unvaccinated individuals, the impact will be to bias estimates towards the null, weakening associations. For other symptoms termed “more exotic” in your comment above, such as skin and subcutaneous outcomes for example, persons are more likely to seek care, and hence capture is more complete and the associations are much stronger. We have included a brief description of this phenomenon in the discussion section (lines 368-369).

Figure 1. Risk of WHO-defined Post-COVID Conditions (PCC) associated with SARS-CoV-2 vaccination status: A sensitivity analysis

Figure 2. Risk of NICE-defined Post-COVID Conditions (PCC) associated with SARS-CoV-2 vaccination status: A sensitivity analysis

I thank the authors for addressing some of my comments. The fact that individual comorbidities are well-balanced between groups is particularly reassuring.

However, I continue to have three concerns:

Comment 1: Poisson regression is unfortunately not less sensitive to competing risks than survival analyses. The risk of death might appear low at the population level, but it can affect different subgroups differently (e.g., pre-omicron, older people, etc.) and might therefore bias subgroup analyses. It is also possible that death is under-recorded, unless the data

were linked to mortality registries.

Response: Thank you for this comment. To address your concerns here around competing risks, we have repeated our primary analysis using Cox cause specific hazard regression time-to-event analysis in which death was considered a competing risk. Compared with the original Poisson model, the results from the Cox regression model are very close (**Figure 3** below), even when restricting the analysis population to older adults aged ≥ 65 years for whom we would expect any bias from competing risk of death to be exacerbated (**Figure 4** below). We do not feel that it is necessary to include this analysis in the appendix, since there are already a number of appendices and we are cautious of over-complicating the paper with statistical language which would divert attention from the primary message.

Figure 3. Risk of Post-COVID Conditions (PCC) associated with COVID-19 vaccination status among persons of all ages before and after using time-to-event analysis (Cox regression): A sensitivity analysis

Figure 4. Risk of Post-COVID Conditions (PCC) associated with COVID-19 vaccination status among adults aged 65+ years before and after using time-to-event analysis (Cox regression): A sensitivity analysis

Comment 2: My original comment on matching for prior influenza vaccination (an important determinant of whether one will receive a COVID-19 vaccine, as mentioned by the authors) seems to have been ignored. Linearly adjusting for such an important covariate is not satisfactory, and I would urge the authors to leverage the high sample size that they have to more tightly control for this important covariate.

Thank you for identifying a slight oversight in our original response. We apologize for missing this request. Influenza vaccination status is mainly used to adjust for healthcare seeking behavior alongside other healthcare utilization data, as we have done in the current analysis. It is highly associated with COVID vaccination status, but theoretically should not be

strongly associated with PCC. Hence, influenza vaccination status is not thought to be a strong confounder in the current analysis, which was our rationale for not matching on this variable in the primary models.

However, as you requested, we have now included an additional sensitivity analysis of individuals matched on influenza vaccine status in the year prior to the date of positive SARS-CoV-2 test. Matching on influenza vaccination status caused a large reduction in the sample size (reducing the sample population down to 46% of the original study population). The results among this influenza vaccination status matched sub-cohort (**Figure 5** below) are similar to the original results, with the exception of associations for renal disorders (effect estimate changed direction [1.03 vs. 0.91], but the conclusion remained unchanged since both confidence intervals of the primary analysis and the sensitivity analysis were wide) and gastrointestinal disorders (point estimates for both analyses were similar [0.97 vs. 0.96], but due to a loss of power in the sensitivity analyses, the association became non-significant). These differences are likely be related to the much smaller sample size and hence random variation. Given the above results, we feel strongly that our decision not to match on influenza vaccination in the main analyses was justified, and we prefer not to include another sensitivity analysis in the manuscript since we would prefer to keep the primary message as clear as possible.

Figure 5. Risk of Post-COVID Conditions (PCC) associated with COVID-19 vaccination status before and after matching on influenza vaccination status in the year prior to the positive SARS-CoV-2 test date: A sensitivity analysis

on how they compare it to references [12] and [16], which both (especially [12]) have enough power to detect associations of interest and both have a 6-month follow-up. I strongly believe that replication of the findings of these studies in this separate and larger sample is worthy of publication, as it is an important research question. However, it should then be made more explicit in the introduction that this is the aim of the study, as I fail to see the novelty otherwise. Of note, reference [14] is unrelated to vaccination and should probably be removed from this passage in the introduction.

Response: Thank you for this comment. We agree that we did not previously emphasize these two recent large-scale studies enough in the original draft or the first revision of our paper. We have now specifically identified these two studies as large EHR-based studies in the introduction that represent a paucity of data on this important research question (line 66). Regarding your comment about reference no. 14, you are correct – this analysis does not consider vaccination status. We have now removed this reference from this sentence as you suggested and have instead included it earlier in the introduction when describing PCC.

Comment 4: The code is not available at that URL, only via direct email to the study authors.

Response: Please note that our analysis uses standard statistical analyses methodology that can be easily replicated using standard statistical programming languages. Further, we have described the statistical methodology in sufficient detail to allow replication of the analysis on alternative EHR-based datasets. However, should readers require SAS codes, the SAS programs used for analysis will be available per request, which is the standard protocol for VSD studies. Many thanks.

Reviewer 3

I would like to thank the authors for addressing my initial comments. The authors have provided a nicely detailed and thorough response to the comments from the previous review and have addressed my major concerns. Following the revision to the article, the authors have sufficiently improved their paper, in reaction to the comments I made.

Response: Thank you for taking the time to review our revisions and for your comments.

In addition to the above comments, we have corrected spelling and grammatical errors, and updated a small number of references. We look forward to hearing from you in due time regarding our submission and to respond to any further questions and comments you may have.

Yours Sincerely,

Debbie E. Malden, DPhil, MSc

(Corresponding author)

Epidemiologist

Kaiser Permanente Southern California

Department of Research & Evaluation

References

Hernandez-Romieu, A. (2020, January 28). Treating Long COVID: Clinician Experience with Post-Acute COVID-19 Care [Presentation slides]. Clinician Outreach and Communication Activity (COCA) Call. Retrieved from <https://www.cdc.gov/me-cfs/pdfs/interagency/treating-long-covid-508.pdf>

Hernandez-Romieu AC, Carton TW, Saydah S, et al. Prevalence of Select New Symptoms and Conditions Among Persons Aged Younger Than 20 Years and 20 Years or Older at 31 to 150 Days After Testing Positive or Negative for SARS-CoV-2. *JAMA Netw Open*. 2022;5(2):e2147053. doi:10.1001/jamanetworkopen.2021.47053

Wanga V, Chevinsky JR, Dimitrov LV, et al. Long-Term Symptoms Among Adults Tested for SARS-CoV-2 — United States, January 2020–April 2021. *MMWR Morb Mortal Wkly Rep* 2021;70:1235–1241.

REVIEWERS' COMMENTS

Reviewer #2 (Remarks to the Author):

I thank the authors for addressing my remaining comments and for conducting several additional analyses to address them. I really think these make the paper stronger.

I would argue in favour of adding the analysis matched on influenza vaccination as a sensitivity analysis. It would not take many words in the methods and results and would provide some evidence that it is not just general vaccine hesitancy which drives the results. The authors mention that influenza vaccination does not influence PCC, but as a proxy for general vaccine hesitancy, I would argue that it does, by virtue of being associated with a range of healthcare use behaviours (and thus both incidence of and diagnosis of several PCC).

Finally, I wonder if the association with increased mental health disorders could be due to the increased stress and anxiety generated by catching COVID-19 despite being vaccinated. If the authors agree, they might want to add this as a potential explanation in the discussion.

April 2024

Ref: Response to reviewer's comments round III (NCOMMS-23-49571B)

Dear Sir/Madam,

Thank you for your time and consideration reviewing our manuscript titled 'Post-COVID conditions following COVID-19 vaccination: A retrospective matched cohort study of patients with SARS-CoV-2 infection' submitted to *Nature Communications*. On behalf of all coauthors, we appreciate your insightful comments on this manuscript, and we have altered our manuscript to reflect these suggestions.

Where appropriate, we have highlighted the changes made in response to your third round of comments in the enclosed revised version of our manuscript. In addition, please find a point-by-point response to specific comments and concerns outlined below in blue font.

Reviewer 2

Overall comment:

I thank the authors for addressing my remaining comments and for conducting several additional analyses to address them. I really think these make the paper stronger.

Response: Thank you for this feedback. We appreciate your time in completing this exceptionally detailed review. We agree that the additional analyses you suggested enhance the study design and increase the reader's confidence in our findings.

Comment 1: I would argue in favor of adding the analysis matched on influenza vaccination as a sensitivity analysis. It would not take many words in the methods and results and would provide some evidence that it is not just general vaccine hesitancy which drives the results. The authors mention that influenza vaccination does not influence PCC, but as a proxy for general vaccine hesitancy, I would argue that it does, by virtue of being associated with a range of healthcare use behaviors (and thus both incidence of and diagnosis of several PCC).

Response: Thank you for this suggestion. We have now included this as an additional sensitivity analysis in the revised version of the manuscript. A brief summary of this analysis is provided in the methods (Lines 399-401) and the results section (154-155). The forest plot is included in the Appendix of the revised manuscript (**Appendix K**).

Comment 2: Finally, I wonder if the association with increased mental health disorders could be due to the increased stress and anxiety generated by catching COVID-19 despite being vaccinated. If the authors agree, they might want to add this as a potential explanation in the discussion.

Response: Thank you for your insightful feedback on this point. We agree that this is an additional potential explanation underlying the association between COVID-19 vaccination and mental health disorders. We are cognizant of over-interpreting the findings, and therefore we have softened the language of your suggested wording slightly: "The observed positive association between COVID-19 vaccination and mental health disorders has not been clearly elucidated. However, in addition to the potential confounding by healthcare seeking behavior described above, the association could be due to the increased stress and anxiety due to individuals experiencing breakthrough SARS-CoV-2 infection despite being vaccinated." (Lines 223-225)

In addition to the above modifications, we have corrected spelling and grammatical errors and added a number of suggested changes from *Nature Communications* editorial team.

Yours Sincerely,

Debbie E. Malden, DPhil, MSc

(Corresponding author)

Epidemiologist

Kaiser Permanente Southern California

Department of Research & Evaluation